# Counterfactual reasoning:
# an analysis of in-context emergence

**Moritz Miller**[12] *   **Bernhard Schölkopf**[12]   **Siyuan Guo**[13]
[1]Max Planck Institute for Intelligent Systems   [2]ETH Zurich   [3] University of Cambridge

## Abstract

Large-scale neural language models exhibit remarkable performance in in-context learning: the ability to learn and reason about the input context on the fly. This work studies in-context counterfactual reasoning in language models, that is, the ability to predict consequences of a hypothetical scenario. We focus on a well-defined, synthetic linear regression task that requires noise abduction. Accurate prediction is based on (1) inferring an unobserved latent concept and (2) copying contextual noise from factual observations. We show that language models are capable of counterfactual reasoning. Further, we enhance existing identifiability results and reduce counterfactual reasoning for a broad class of functions to a transformation on in-context observations. In Transformers, we find that self-attention, model depth and pre-training data diversity drive performance. Moreover, we provide mechanistic evidence that the latent concept is linearly represented in the residual stream and we introduce designated *noise abduction heads* central to performing counterfactual reasoning. Lastly, our findings extend to counterfactual reasoning under SDE dynamics and reflect that Transformers can perform noise abduction on sequential data, providing preliminary evidence on the potential for counterfactual story generation. Our code is available under `https://github.com/mrtzmllr/iccr`.

## 1   Introduction

> Thinking is acting in an imagined space. – Konrad Lorenz [Lorenz, 1973]

Large language models demonstrate remarkable capability in task-agnostic, few-shot performance, such as in-context learning and algorithmic reasoning [Brown et al., 2020]. Despite their success, one of the grand challenges of artificial general intelligence remains the ability to unearth novel knowledge. This requires principled reasoning over factual observations instead of inventing information when uncertain, a phenomenon termed hallucination [Achiam et al., 2023].

Thinking, in Konrad Lorenz's words, is acting in an imagined space. Counterfactual imagination/reasoning, studied in early childhood cognition development [Southgate and Vernetti, 2014] and formalized in causality [Schölkopf, 2022, Pearl, 2009, Hernan, 2024], predicts the consequences of changes in hypothetical scenarios. It answers "what if" questions and predicts potential outcomes that could have occurred had different actions been taken. Such principled thinking enables *fast* and *responsible* learning, e.g., efficient reinforcement learning [Mesnard et al., 2021, Lu et al., 2020], counterfactual generative networks [Sauer and Geiger, 2021], responsible decision-making in complex systems [Wachter et al., 2017], and fairness [Kusner et al., 2017].

In precision healthcare, one may ask what would have happened had a patient not received treatment to assess the individualized treatment effect. One step further, *in-context* counterfactual reasoning pertains the hypothetical consequence for an individual after observing the effect of different treatments

---

*Correspondence to `moritz.miller@tuebingen.mpg.de`

39th Conference on Neural Information Processing Systems (NeurIPS 2025).

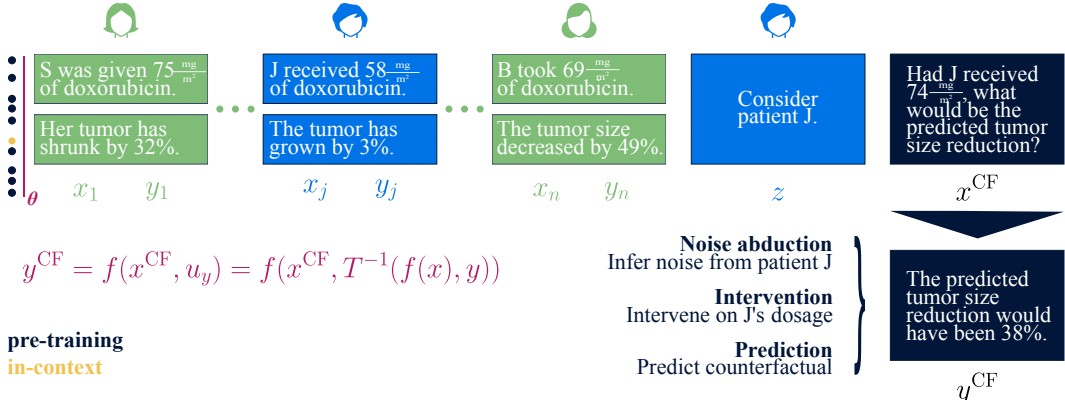

Figure 1: **In-context counterfactual reasoning.** Training on a corpus of sequences that come from a mixture of distributions (each ● on the far left represents a single sequence from a distinct distribution parameterized by $\theta$). Suppose each observation satisfies $y = f(x, u_y)$ for some noise $u_y$. An in-context sequence ● takes the form of $n$ examples. This is concatenated with index token $z$ referring back to observed factual observation $(x_j, y_j)$ when $z = j$ and the hypothetical new information $x^{\text{CF}}$: $(x_1, y_1, \ldots, x_n, y_n, z, x^{\text{CF}})$. In-context counterfactual reasoning can be measured via accurate prediction on $y^{\text{CF}}$. Accurate prediction requires *noise abduction* from factual observation, that is, to infer $u_y$ consistent with $(x_j, y_j)$, and *prediction* based on the *intervention* $x^{\text{CF}}$ and inferred $u_y$.

on multiple patients. Figure 1 displays a simplified setting where $n$ patients receive neoadjuvant chemotherapy to treat breast cancer. Given different dosages of doxorubicin, the tumor size reduction is recorded as output. A counterfactual reasoner now imagines the tumor size reduction of patient $j$, had she received $45\frac{\text{mg}}{\text{m}^2}$ of doxorubicin instead of the prescribed dosage of $65\frac{\text{mg}}{\text{m}^2}$. Augmented with agentic verification, such principled counterfactual reasoning would be a valuable tool for automatic scientific discovery and a guardrail for safe AI deployment.

To concretely understand to what extent language models perform in-context counterfactual reasoning, we use the counterfactual framework of Pearl [2009] and study a controlled synthetic setup similar to Garg et al. [2022]. That is, let $y = f(x, u_y)$ for some function $f \in \mathcal{F}$, for function class $\mathcal{F}$. The model predicts target $y^{\text{CF}}$ given counterfactual query $x^{\text{CF}}$ conditioned on a prompt sequence $(x_1, y_1, \ldots, x_k, y_k, z, x^{\text{CF}})$ where $z$ is an index token indicating the position of the factual observation that such counterfactual query is based on. Given factual observation $(x, y)$, counterfactual reasoning requires three steps:

- noise abduction: infer $u_y$ that is consistent with the factual observation $(x, y)$,

- intervention $do(X = x^{\text{CF}})$: intervene on $X$ to consider the hypothetical value $x^{\text{CF}}$,

- prediction $y^{\text{CF}} = f(x^{\text{CF}}, u_y)$: predict the effect of changes $x^{\text{CF}}$ by inheriting noise $u_y$ from the factual observation.

For a complete introduction on the basics of causality, see Section 2.1. In contrast to natural language, under this controlled setup, we can formally ask

*Can language models perform in-context counterfactual reasoning?*

with concrete metrics $\mathbb{E}\left[\ell(y^{\text{CF}}, y^{\text{CF}}_{\text{pred}})\right]$, for model prediction $y^{\text{CF}}_{\text{pred}}$ and squared error function $\ell$.

This work studies counterfactual reasoning in exchangeable data, as we note language models are pre-trained on a mixture of sequences coming from different distributions. The training dynamics allow random permutations over different input sequences, i.e., implicitly assume sequences are exchangeable (see Def. 1). De Finetti [1931] shows that any exchangeable sequence can be modeled as a mixture of conditionally i.i.d. sequences. In other words, there exists a latent variable $\theta$ with

probability measure $\pi$ such that

$$P(s_1, \ldots, s_n) = \int \prod_{i=1}^n p(s_i \mid \theta) d\pi(\theta) \tag{1}$$

for $s_i$ the $i^{\text{th}}$ sequence. Pre-training thus amounts to learning mutual information $\theta$ between sequences and their prior $\pi(\theta)$. At inference time, the model computes the *posterior predictive distribution*: given a sequence $s_i$ with tokens $x_1^i, \ldots, x_t^i$, the next token prediction writes

$$p(x_t^i | x_{<t}^i) = \int_\theta p(x_t^i | x_{<t}^i, \theta) \pi(\theta | x_{<t}^i) \mathrm{d}\theta.$$

Guo et al. [2024a,b] have established a theoretical foundation to study observational and interventional causality in exchangeable data. We extend this framework to counterfactuals. Building upon previous work, our contributions are:

- We relate in-context counterfactual reasoning to transformations on the factual in-context observations (Lemma 1). Extending known identifiability results [Nasr-Esfahany et al., 2023], we provide theoretical guarantees for our exchangeable setup (Theorem 1).
- We empirically show that in-context counterfactual reasoning emerges in Transformers in form of a designated *noise abduction head* (Section 4.4). Causal probing reveals that the residual stream linearly encodes the latent $\theta$ (Section 4.2).
- We find that data diversity in pre-training, self-attention and model depth are key for Transformers' performance (Sections 4.2 and 4.3). More interestingly, our findings transfer to cyclic sequential data (Section 5), demonstrating concrete preliminary evidence that language models can perform counterfactual story generation in sequential data.

## 2 Background

### 2.1 Exchangeability and causality

**Definition 1** (Exchangeable sequence). A sequence of random variables is exchangeable if for any finite permutation $\sigma$ of its indices,

$$\mathbb{P}(S_1, \ldots, S_n) = \mathbb{P}(S_{\sigma(1)}, \ldots, S_{\sigma(n)}).$$

Here we treat each random variable $S_i$ as a sequence with bounded context length $t$.

**Causality fundamentals.** When $X$ causes $Y$, the structural causal model (SCM) is determined by functional assignments $f_X, f_Y$ and exogenous variables $U_X, U_Y$, i.e., $X := f_X(U_X), Y := f_Y(X, U_Y)$. An intervention performed on variable $X$ is represented as $\text{do}(X = x)$. Thus, one replaces $X$ with value $x$ and allows $Y$ to inherit the replaced value, i.e., $X := x, Y := f_Y(x, U_Y)$. A *counterfactual statement* of the form "$Y$ would be $y$ had $X$ been $x$ in situation $U = u$" is often denoted as $Y_x(u) = y$. For explicit representation, we use $y^{\text{CF}}, x^{\text{CF}}$ for values intended for counterfactual predictions. In contrast to the interventional setting, the counterfactual entity inherits noise $u_y$ consistent with the factual observation.

### 2.2 Transformer architecture

Transformers [Vaswani et al., 2017] operate on a sequence of input embeddings by passing them to blocks consisting of attention and a multi-layer perceptron (MLP). Mimicking present-day large language models, this work focuses on the decoder-only, autoregressive GPT-2 [Radford et al., 2019] architecture for next-token prediction. The softmax operation with causal masking is denoted by $\text{softmax}_*$. Given a sequence of input embeddings $\mathbf{E} \in \mathbb{R}^{T \times E}$ of context length $T$ and embedding dimension $E$, the model first projects each token into $D$-dimensional hidden embeddings via $\mathbf{X}_0 = \mathbf{E}W_E + \text{pos}(\mathbf{E})W_P$, for embedding matrix $W_E \in \mathbb{R}^{E \times D}$, positional embedding $W_P \in \mathbb{R}^{T \times D}$ with absolute positional encoding. Given an input sequence of embeddings, multi-head attention at layer $l$ passes the embeddings to query, key, value weight matrices $W_Q^h, W_K^h, W_V^h \in \mathbb{R}^{D \times D}$ and computes attention per head as $\mathbf{A}_l^h = \text{softmax}_* \left( \mathbf{X}_{l-1}W_Q^h (\mathbf{X}_{l-1}W_K^h)^\top \right)$. Then the model

concatenates the multi-head output and transforms it via output weight matrix $W_O \in \mathbb{R}^{H \cdot D \times D}$ as $\mathbf{M}_l = \text{Concat}(\mathbf{A}_l^1, ..., \mathbf{A}_l^H)W_O$. The output is added into the residual stream as $\mathbf{R}_l = \mathbf{X}_{l-1} + \mathbf{M}_l$ and passed to an MLP $\mathbf{X}_l = \text{MLP}(\mathbf{R}_l) + \mathbf{R}_l$. After the last Transformer layer, the embeddings are mapped back to logits through the unembedding matrix: $\mathbf{O} = \mathbf{X}_L W_U$. We ignore layer normalization.

## 3 In-context counterfactual reasoning

We study a linear regression task that requires noise abduction. The structural causal model (SCM) considered is $f_X(U_X) := U_X, f_Y(X, U_Y) := \beta X + U_Y$, where $U_X, U_Y$ are independent exogenous variables. We are interested in accurate counterfactual prediction on $y^{\text{CF}}$ given new information $x^{\text{CF}}$ and factual observation $(x, y) = (u_x, \beta u_x + u_y)$, where $y^{\text{CF}} = \beta x^{\text{CF}} + u_y$.

**Pre-training sequence generation.** Our pre-training corpus consists of a mixture of sequences coming from different distributions that share the same causal structure but different functional classes. Throughout, we use the terms *training* and *pre-training* interchangeably. To generate a single sequence, we randomly draw a latent variable $\theta$ and parameterize both the regression coefficient $\beta$ and noise variables $U_X, U_Y$ based on $\theta$. Each sequence takes the form $(x_1, y_1, \ldots, x_{n_i}, y_{n_i}, z, x^{\text{CF}}, y^{\text{CF}})$, where $n_i$ in-context examples are sampled from the SCM parameterized via $\theta$ and $z$ denotes the index token for that factual observation which the counterfactual query would be based on. We vary the number of in-context examples $n_i$ observed at each sequence to avoid prediction based on information from positional encoding only. In particular, we respect the causal attention mask by ordering observed variables in topological order in a sequence format.

**Training objective.** We minimize $\mathbb{E}\left[\ell(y^{\text{CF}}, y_{\text{pred}}^{\text{CF}})\right]$, for $y_{\text{pred}}^{\text{CF}}$ the model's predicted outputs and $y^{\text{CF}}$ the true counterfactual completion. We choose $\ell$ to be the mean squared error (MSE).

In-context counterfactual reasoning can be represented via the *posterior predictive distribution*

$$p(y^{\text{CF}}|x_1, y_1, \ldots, x_n, y_n, z, x^{\text{CF}}) = \int_\Theta \int_B \delta(Y^{\text{CF}} = \beta x^{\text{CF}} + y_z - \beta x_z)p_\beta(\beta|\mathbf{x}, \theta)\pi(\theta|\mathbf{x})\mathrm{d}\beta\mathrm{d}\theta. \quad (2)$$

The model is required to learn how to compute posteriors $p_\beta, \pi$ given a query at pre-training. In-context completion reduces to identifying $z$, inserting query elements $(x_z, y_z, x^{\text{CF}})$ and weighting the resulting $y_z^{\text{CF}}$ with the posterior of $\beta$ given $\mathbf{x}$.

We first present Lemma 1 and show how accurate prediction on counterfactual values $y^{\text{CF}}$ can be reduced to a transformation on in-context observations. Lemma 1 incorporates a broad class of functions, including additive noise models (ANMs) [Peters et al., 2011, 2014], multiplicative noise models and exponential noise models.

**Lemma 1** (Counterfactual reasoning as transformation on observed values). Suppose $y = T(f(x), u)$ for some function $T : \mathcal{Y} \times \mathcal{U} \longrightarrow \mathcal{Y}$. Assume for any fixed $f(x) \in \mathcal{Y}$, the inverse $T^{-1}(f(x), \cdot)$ exists for all $y$, i.e., $u = T^{-1}(f(x), y)$. Then, counterfactual reasoning reduces to learning a transformation $h$ on observed factual observations $(x, y)$ and counterfactual information $x^{\text{CF}}$ with

$$y^{\text{CF}} = h\left(f(x^{\text{CF}}), f(x), y\right), \quad (3)$$

where $h(f(x^{\text{CF}}), f(x), y) = T(f(x^{\text{CF}}), T^{-1}(f(x), y))$ operates on elements of $\mathcal{Y}$ only.

Lemma 1 shows that given the existing information in the query, in-context counterfactual reasoning is no more than estimating the transformed function $T, f, T^{-1}$ on the fly – a task Transformers are known to be capable of performing [Garg et al., 2022, Akyürek et al., 2023, von Oswald et al., 2023]. The assumptions in Lemma 1 cover a broad class of functions, for example,

- **Additive noise models (ANMs).** Functions taking the form $Y = f(X) + U$ for arbitrary function $f$ can be equivalently represented as $y := T(f(x), u) = f(x) + u$. For fixed $f(x)$, the inverse $T^{-1}(f(x), \cdot)$ exists and reads $u = T^{-1}(f(x), y) = y - f(x)$.

- **Multiplicative noise models.** Functions taking the form $Y = f(X) \cdot U$ for arbitrary function $f$. For fixed $f(x)$, the inverse $T^{-1}$ exists and reads $u = T^{-1}(f(x), y) = \frac{y}{f(x)}$.

- **Exponential noise models.** Functions taking the form $Y = \exp(f(X) + U)$ with inverse $u = T^{-1}(f(x), y) = \log(y) - f(x)$.

On top of that, Nasr-Esfahany et al. [2023] provide three cases under which the class of bijective generation mechanisms (BGMs) is counterfactually identifiable. Under a set of assumptions, one can disentangle the causal mechanisms solely by knowing the causal graph and observational data. Our setup can be subsumed under that framework of BGMs. We therefore extend the result in the Markovian case [Nasr-Esfahany et al., 2023] to our exchangeable setup and show that the transformation on observed values is counterfactually identifiable. Details are deferred to Appendix B.

**Theorem 1** (Counterfactual identifiability under exchangeability). *Let $X, Y, U, \theta$ scalar random variables with $X \perp\!\!\!\perp U | \theta$. Take $T : \mathcal{Y} \times \mathcal{U} \longrightarrow \mathcal{Y}$ with $y = T(f(x), u)$. Assume $\forall f(x) \in \mathcal{Y}$, the inverse $T^{-1}(f(x), \cdot)$ exists for all $y$, i.e., $u = T^{-1}(f(x), y)$, and $\forall f(x) \in \mathcal{Y}$, $T(f(x), \cdot)$ is continuous. Suppose further that $f(x)$ continuous $\forall x$, and strictly monotonic in $x$. Then, set $Y = T(f(X), U)$. Given the joint law $\mathbb{P}_{X,Y|\theta}$, $T$ is counterfactually identifiable.*

## 4 Experiments

Based on the training setup described in Section 3 with details in Appendix C, we choose a controlled synthetic setup to allow concrete evaluation on in-context counterfactual reasoning.

### 4.1 Language models can in context perform counterfactual reasoning on linear functions

We assess the impact of model architecture choice on in-context counterfactual reasoning performance. In contrast to natural language tasks, the performance in our task is clearly defined and can be measured as:

- *accuracy* in prediction: $\text{MSE}(y^{\text{CF}}, \widehat{y^{\text{CF}}})$,
- *learning efficiency*: number of in-context examples seen to achieve satisfactory prediction.

We compare the STANDARD decoder-only GPT-2 Transformer architecture against several autoregressive baselines. We use the terms STANDARD and GPT-2 interchangeably. The models considered are:

- GPT-2 [Radford et al., 2019]: 12 layers, 8 attention heads, hidden dimension 256,
- LSTM [Hochreiter and Schmidhuber, 1997]: 2 layers, hidden dimension 256,
- GRU [Cho et al., 2014]: 2 layers, hidden dimension 256,
- ELMAN RNN [Elman, 1990]: 2 layers, hidden dimension 256.

We evaluate each model on unseen sequences sampled in-distribution and report results averaged over 6400 sequences. Our synthetic generation yields $\mathbb{E}[Y^{\text{CF}}] = 0$ and $\log(\text{std}(Y^{\text{CF}})) = 2.56$. Appendices C and D include data generation particularities as well as experimental and model details.

Figure 2 compares the impact of different model architectures on in-context counterfactual reasoning. We plot the log-transformed in-context MSE against the number of in-context examples observed in a prompt. We hypothesize that more observed examples lead to higher counterfactual prediction accuracy, and that better models require less in-context examples for accurate prediction. For contexts of over 14 in-context examples, the Elman RNN achieves significantly higher in-context MSE than the rest. Across LSTM, GRU and STANDARD Transformer, we observe no significant performance difference. In Appendix D, we reference literature discussing the RNN architectures. For now, we confine ourselves to autoregressive Transformers such that all future statements refer to this setup only.

### 4.2 Counterfactual reasoning emerges in self-attention

**Self-attention.** Lemma 1 shows the reliance of counterfactual reasoning on copying in-context observed values. Self-attention has been shown to perform copying by implementing induction heads [Olsson et al., 2022, Akyürek et al., 2024]. We thus hypothesize that counterfactual reasoning emerges in the attention sublayers of the Transformer.

We conduct an experiment that switches on and off attention layers in the Transformer and separately train each model on our task. Figure 3a shows the in-context MSE for **MLP-Only** and **AO** (attention-only) with 2, 4, 8 layers and the STANDARD setup. At training time, we observe that **AO** models mimic the loss curve of the STANDARD setup, while the **MLP-Only**'s loss remains constant at

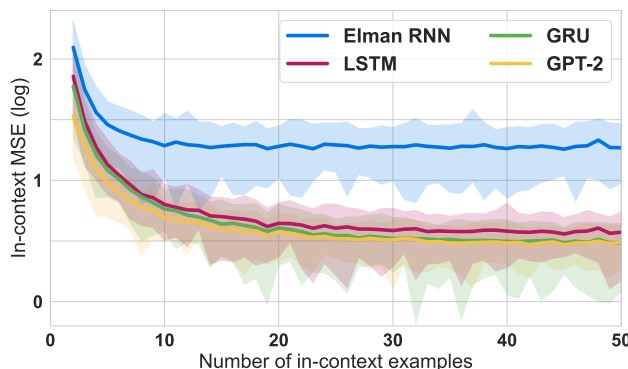

Figure 2: **Model comparisons for in-context counterfactual reasoning.** In-context counterfactual prediction accuracy measured via log-transformed MSE averaged over 6400 sequences versus the number of in-context examples observed in a prompt. We compare GPT-2 (STANDARD), LSTM, GRU, and Elman RNN. Though not significant, GPT-2 achieves lowest error and fastest convergence rate for a small number of in-context examples. For more than 14 in-context examples, the Elman RNN obtains significantly [Efron, 1979] higher in-context MSE than the three other architectures.

the level of the theoretical variance, $\mathrm{Var}(Y^{\mathrm{CF}})$. Figure 3a shows that counterfactual reasoning performance improves for all considered Transformers with increasing context length. Although at a higher loss than the STANDARD Transformer, smaller models do perform better on longer contexts. Oblivious of context length, **MLP-Only** does not appear to reason counterfactually.

**Model depth.** Elhage et al. [2021] show that self-attention performs a copying task. Lemma 1 relates counterfactual reasoning to copying multiple values and learning transformations. Such a task requires the composition of attention heads to pass information across layers. We hypothesize that model depth plays an important role. Fixing the total number of attention heads at 8, we train 4 Transformer-based models with increasing depth: from 1-layer, 8-head Transformers to models with 8 layers of 1 head each. We expect that if depth does not influence performance, given the same number of attention heads, all models should have comparable in-context MSE. Figure 3b represents the loss on 6400 sequences of 35 in-context examples for both **Full** and **AO** models. Additional results can be found in appendix D. With increasing depth, loss declines for both setups. In particular, the **Full** 8-layer, 1-head Transformer yields lowest MSE, while the 4-layer, 2-head designs achieve competitive results.

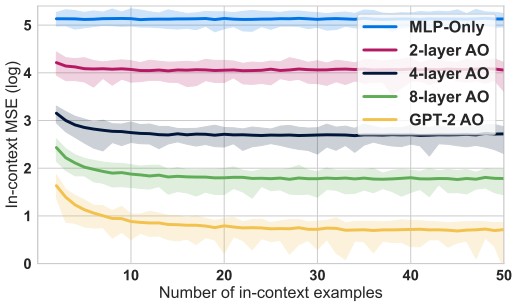

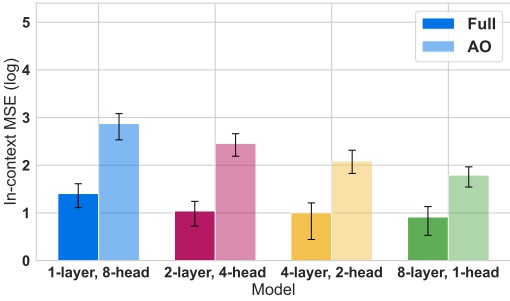

(a) **Switching attention.** In-context MSE stagnates with increasing number of in-context examples observed for the **MLP-Only** model. We compare to attention-only (**AO**) Transformers of 2, 4, 8 layers with 1 head each as well as the STANDARD setup. For all Transformers, in-context MSE decreases as more in-context examples are observed.

(b) **Varying depth.** Keeping the number of attention heads constant at 8, we compare **Full** and attention-only (**AO**) Transformers with 1 layer, 8 heads; 2 layers, 4 heads; 4 layers, 2 heads; 8 layers, 1 head. We observe a decrease in in-context loss as model depth increases. We evaluate on 6400 sequences of 35 in-context examples each.

Figure 3: **Attention and model depth matter.**

**Latent parameter.** A final way to confirm the relevance of self-attention is by probing the residual stream for the latent $\theta$. As $\theta$ governs the distribution of the in-context sequence, it is natural to check for that signal. By doing so, we find that the 8-layer **AO** Transformer learns the latent parameter early on. Figure 4a shows that a linear model predicting $\theta$ from the residual stream after layer 2 does so at adjusted $R^2 \approx 0.9104$. After this initial spike, all future layers absorb that information as the adjusted $R^2$ stays above 0.9. In addition we train separate probes on the regression parameter $\beta$ and find similar results to probing for $\theta$. As $\beta \sim \mathcal{N}(\theta, 1)$, this is in line with our intuition. Moreover, Figure 4b checks whether the difference of residual streams after and before every layer carries latent information. In fact, we have additional evidence that layers 2 and 5 are central in learning $\theta$ (and $\beta$), while the final layer's contribution is marginal at adjusted $R^2 < 0.2$. By running this standard experiment, we find that self-attention alone suffices to linearly encode the latent theme. We thus provide additional support for the Linear Representation Hypothesis [Park et al., 2024].

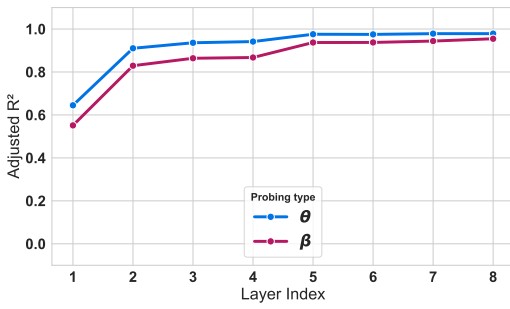
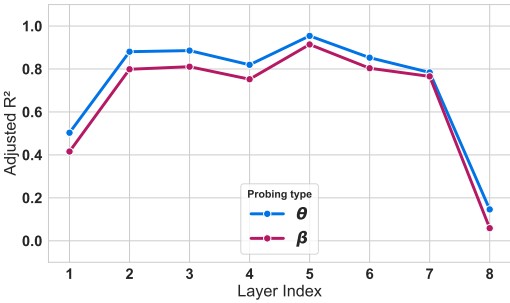

(a) **Relevance at every layer.** We plot the adjusted $R^2$ after every layer against the layer indices. Starting at the second layer, we observe adjusted $R^2 > 0.9$ for predicting $\theta$, indicating that the latent is encoded linearly in the residual stream.

(b) **Additional relevance of this layer.** We subtract the residual streams before each layer from the residual stream after and report the adjusted $R^2$. Layers 2 and 5 are especially relevant while layer 8 does not add substantial information.

Figure 4: **The Transformer linearly encodes the latent parameter.** We train a linear probe on 6400 prompts from a fresh evaluation set after every layer. We evaluate on 1280 sequences. All layers after the first one encode relevant information for predicting $\theta$ from the residual stream only.

### 4.3 Data diversity and out-of-distribution (*OOD*) generalization

Causal structure identification, a previously deemed impossible task from i.i.d. observational data alone [Pearl, 2009], has been shown to be feasible in multi-domain data [Guo et al., 2024a], a natural setting for exchangeable data. We hypothesize that (counterfactual) reasoning is similarly only feasible under diverse pre-training data. As a measure of pre-training data diversity, we use the effective support size [Grendar, 2006], defined as

$$\text{Ess}(\theta) := \exp\left(-\sum_{\vartheta \in \Theta_0} p(\vartheta) \log p(\vartheta)\right) = \exp\left(H(\theta)\right),$$

where $\Theta_0$ is the space of latent $\theta$ sampled at pre-training, and $H$ the Shannon entropy [Shannon, 1948]. We further compare the pre-training data diversity's impact on simple *OOD* generalization. We pre-train on both UNIFORM and NORMAL sampling of latent $\theta$ and evaluate on UNIFORM sampled test data. Figure 5a shows a clear trend that diverse pre-training data, measured as a higher effective support size, yields lower in-context MSE for both in-distribution and *OOD* cases. Appendix E shows that such generalization ability persists when evaluated on NORMAL, and that in-context loss declines with increasing data diversity across a changing number of in-context examples. We also discuss how this relates to existing research.

### 4.4 Emergent model behavior

The ordinary least squares estimator for the regression parameter $\beta$ writes

$$\hat{\beta} = \frac{\sum_{i=1}^{n}(x_i - \bar{x})(y_i - \bar{y})}{\sum_{i=1}^{n}(x_i - \bar{x})^2}$$

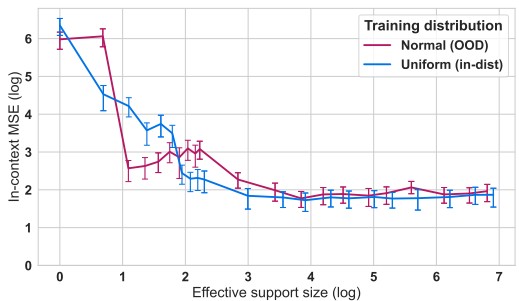 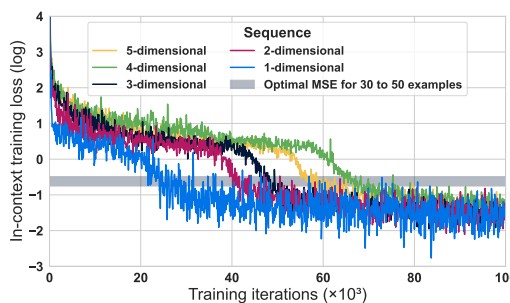

(a) **Data diversity in pre-training.** In-context MSE (log-scaled) averaged over 6400 prompts at 35 in-context examples against log-scaled effective support size. Each point represents one fully pre-trained model on either the UNIFORM sampled or NORMAL sampled $\theta$. Models are evaluated on UNIFORM.

(b) **Phase transitions during training.** In-context training MSE against the number of training steps. When training on $100'000$ iterations, we observe that the model with $E = 5$ reaches a phase transition after more than $50'000$ steps. Depending on embedding dimension $E$, the phase transition occurs sooner or later.

Figure 5: **Data diversity and emergence.** In-context counterfactual reasoning *emerges* at training. To generalize to unknown $\theta$, the model is to be trained on a sufficiently diverse pre-training corpus.

which leads to the fitted value $y_{\hat{\beta}}^{\text{CF}} = \hat{\beta}(x^{\text{CF}} - x_z) + y_z$, and the baseline MSE between the linear regression implementation $y_{\hat{\beta}}^{\text{CF}}$ and the ground truth counterfactual, $\text{MSE}\left(y_{\hat{\beta}}^{\text{CF}}, y^{\text{CF}}\right)$. We refer to this as the *(estimated) optimal* MSE and note that it lies between $0.6132$ and $0.4739$ for 30 to 50 in-context examples. We therefore compare models trained on data with embedding dimension $E \in \{1, 2, 3, 4, 5\}$ to this estimated optimal MSE. Here, we fix $z = 14$ to enable faster learning relative to the setup with variable index token. Figure 5b displays the training progression for **Full** GPT-2 Transformers trained for $100'000$ steps. We observe that the ability to attain the estimated optimal MSE *emerges* as training proceeds [Wei et al., 2022, Nanda et al., 2023]. Given that we primarily train on $50'000$ training steps, we observe that at $E = 5$, the counterfactual reasoning behavior is yet to emerge. For sufficient training length, therefore, the model implements a solution performing better than the estimated optimal MSE. In Appendix E, we confirm this behavior for variable $z$ by introducing a designated *noise abduction head* which emerges at the 7th layer of the **Full** GPT-2 architecture. We also move beyond linear regression, and we compare our results to models trained on the setup in Garg et al. [2022].

## 5  Cyclic sequential dynamical systems

Language is sequential. Sentences within a story depend on each other. Counterfactual story generation naturally occurs by prompting the model with a factual story and querying it to complete the story under a hypothetical scenario. This is done while keeping the noise unchanged. Although difficult to evaluate in language, our key insight is that such behavior can be mimicked by ordinary differential equations modeling the underlying causal mechanism [Mooij et al., 2013, Peters et al., 2022, Lorch et al., 2024]. Given an initial condition $(x_0, y_0)$, a dynamical system determines the state $(x_t, y_t)$, for $t \geq 0$. We model causal dependencies using Itô stochastic differential equations (SDEs),

$$
\begin{aligned}
\mathrm{d}Y_t &:= f(X_t, Y_t)\mathrm{d}t + \sigma_Y \cdot \mathrm{d}W_t \\
\mathrm{d}X_t &:= g(X_t, Y_t)\mathrm{d}t + \sigma_X \cdot \mathrm{d}U_t \\
X_0 &:= \xi_0 \text{ and } Y_0 := \upsilon_0,
\end{aligned}
\tag{4}
$$

for initial condition $(\xi_0, \upsilon_0)$. Here, $W_t, U_t$ denote independent Brownian motions with constant diffusion by $\sigma_X, \sigma_Y$, and $f, g$ represent the drift coefficients for $X, Y$, respectively.

Autoregressive models are discrete. To address the challenge of adapting continuous SDE data generation to discrete observations, we draw $n$ event times from a uniform distribution over a bounded interval. We evaluate the above SDE at time $t_m$ for all $m \leq n$ and input observations to the model. Instead of deterministically slicing the interval into equivalent regions, we can capture the complete bounded interval as the randomly sampled set of event times, independent across iterations. Noise abduction is thus required over both the realization of $u$ and of event times $t_m \leq t_n$.

In summary, we provide context $(x_{t_1}, y_{t_1}, ..., x_{t_n}, y_{t_n})$, a counterfactual token indicating the start of counterfactual story generation, and the start of the counterfactual sequence $(x_{t_1}^{\text{CF}}, y_{t_1}^{\text{CF}})$. Then, we train the model on completing the full counterfactual story. We thus ask the model to generate $(x_{t_2}^{\text{CF}}, y_{t_2}^{\text{CF}}, ..., x_{t_n}^{\text{CF}}, y_{t_n}^{\text{CF}})$. Models are evaluated on the MSE across all predicted tokens. Contrary to above, we assess performance not just on the final token, but over the entire counterfactual continuation consisting of $(n-1) \cdot 2$ tokens. Additionally, the full context corresponds to a *single* instance of a factual story. Given this story, the model must infer the noise associated with all observational examples, rather than just a single example pair. We thus have deterministic functional assignments $f, g$, and copy the noise $W_t, U_t$ from the observational sequence. With constant diffusion, we invoke Lemma 1, which connects the cyclic extension to the regression setup,

$$Y_t^{\text{CF}} = \int_0^t f(X_s^{\text{CF}}, Y_s^{\text{CF}})\mathrm{d}s - \int_0^t f(X_s, Y_s)\mathrm{d}s + Y_t. \tag{5}$$

We conduct preliminary experiments on data generated from the Lotka-Volterra model [Lotka, 1910] describing predator-prey dynamics,

$$f(X_t, Y_t) := \alpha X_t - \beta X_t Y_t \tag{6}$$
$$g(X_t, Y_t) := -\gamma Y_t + \delta X_t Y_t \tag{7}$$

with $\alpha, \beta, \gamma, \delta \geq 0$. Appendix F specifies training and data generation details and analyzes the attention behavior in the Transformer. Figure 6a shows that the **Full** GPT-2 Transformer obtains the lowest in-context MSE across the four studied models. Error bars [Efron, 1979] point to competitive performance across RNN-type architectures. We evaluate the models on contexts of $n = 20$ realizations of in-context pairs. Recall that these are not *conditionally i.i.d.* in-context examples as above, but rather follow a sequential pattern. Figure 6b illustrates this sequentiality. Given a query consisting of interleaved observations of the prey, predator concentrations $(x, y)$, the model infers the underlying latent $\theta$ governing the system. It then infers the full counterfactual trajectory beyond $(x_{t_1}^{\text{CF}}, y_{t_1}^{\text{CF}})$. Notably, while the observed $y \in [1.1, 1.9]$, the model correctly infers that the counterfactual is realized at a lower level, $y^{\text{CF}} \in [0.1, 0.7]$. This indicates that the model internalizes the underlying dynamics and uses them to generate coherent predictions.

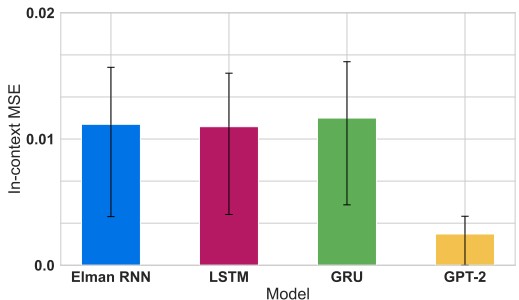
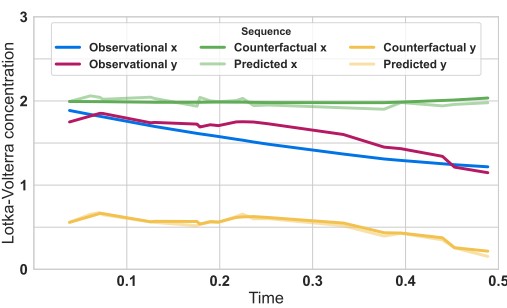

(a) **In-context evaluation on Lotka-Volterra SDEs.** In-context MSE of four models queried with data on 20 distinct time points. The MSE is competitive for Elman RNN, LSTM, and GRU at over 0.1. For the **Full** GPT-2 Transformer, we observe MSE $\approx 0.0025$, which is significantly below the performance of the former three architectures.

(b) **Prediction of Lotka-Volterra SDEs in-context.** The query consists of the observational interwoven $(x, y)$ data. In-context, the model learns the dynamics governed by the latent $\theta$ and completes the counterfactual trajectory for all $t_m > 0$. Although observational $y$ spans $[1.1, 1.9]$, the model infers that counterfactual $y^{\text{CF}}$ is constrained to $[0.1, 0.7]$.

Figure 6: **Cyclic causal relationship.**

## 6    Discussion and related work

**Counterfactual reasoning in natural language.** Prior investigation into *counterfactual story generation* has been mainly focused on empirical evaluation on off-the-shelf language models. Tandon et al. [2019], Li et al. [2023] study lexical associations between factual statements and the hypothetical *"what if"* scenario; Qin et al. [2019] provide a dataset for counterfactual rewriting

evaluation. Bottou and Schölkopf [2025] link the ability to hypothesize about counterfactuals to generating meaningful text beyond the training distribution. Ravfogel et al. [2025] study language directly as structural causal acyclic models and Yan et al. [2023] focus on representation identification for adaptive counterfactual generation. This work respects the sequential nature of language and performs controlled pre-training and concrete evaluations that demonstrate preliminary and promising evidence that language models are capable of counterfactual story generation.

**In-context learning.** Language models [Brown et al., 2020] have shown surprising in-context reasoning ability demonstrating the potential for practical user interactions in the form of chatbots. Transformers, the backbone architecture for modern language models, have been shown to in-context learn different function classes [Garg et al., 2022]. Some studies investigate the models' ability to perform gradient descent over in-context examples [Dai et al., 2023, Deutch et al., 2024, von Oswald et al., 2023] while others discuss implicit Bayesian inference [Xie et al., 2022, Falck et al., 2024, Ye and Namkoong, 2024] and *induction heads* [Olsson et al., 2022, Akyürek et al., 2024]. A recent line of work considers guarantees for in-context learning algorithms on linear models [Akyürek et al., 2023].

**Causality and exchangeability.** Pearl [2009], Hernan [2024] formalize the notion of counterfactuals in causality. Peters et al. [2022], Lorch et al. [2024] study cyclic causal models via differential equations. Mixture of i.i.d. data or multi-domain data have been the foundation of modern pre-training corpora. Guo et al. [2024a] demonstrate that inferring causality is feasible in such a mixture of i.i.d. data, providing the potential for language models to exhibit causal reasoning capabilities. Recent work has continued to investigate connections between causality and exchangeability from intervention [Guo et al., 2024b] to representation learning [Reizinger et al., 2025]. Concurrently, interest in exchangeability and language models has resurfaced. Zhang et al. [2023] study topic recovery from language models and Falck et al. [2024], Ye and Namkoong [2024], Xie et al. [2022] discuss in-context learning as performing implicit Bayesian inference in the manner of de Finetti. This provides potential for *causal* de Finetti, which allows structured in-context learning with the potential for principled reasoning and fast inference. This work serves as a preliminary study on in-context counterfactual reasoning in the bivariate case and demonstrates its potential.

**Limitations.** Our study is focused on controlled synthetic datasets to circumvent evaluation difficulties inherent to natural language. We therefore assess reasoning from a causal perspective [Peters et al., 2017, Mondorf and Plank, 2024] as algebraic manipulation of existing knowledge [Bottou, 2014]. We only train on small-scale GPT-2 type models to demonstrate preliminary evidence on counterfactual story generation. Furthermore, we assume no observed confounders, which is difficult to satisfy in the real world, as observations are finite and contain missing information which requires *out-of-variable* generalization [Guo et al., 2023]. Realistic benchmarks on natural language evaluation and larger-scale pre-training that provide principled reasoning for real-world impact, such as scientific reasoning, represent potential avenues for future research.

**Broader impacts.** Since our models are not trained on natural language, the typical societal concerns associated with counterfactual story generation do not directly apply. We view our current work as a scientific exploration of desirable properties for the next generation of AI models.

# 7 Conclusion

We discuss in-context counterfactual reasoning in both linear regression tasks and cyclic sequential data. We observe that counterfactual reasoning can emerge in language models. We find that Transformers, in particular, rely on self-attention and model depth for in-context counterfactual reasoning. Data diversity in pre-training is essential for the emergence of reasoning and *OOD* generalization. We provide insights on how counterfactual reasoning can be interpreted as a copying and function estimation task, with theoretical and empirical evidence over a broad class of invertible functions (1). We mechanistically support our findings by introducing a designated *noise abduction head* and probing the residual stream for the latent concept. We develop a sequential extension grounded in the Lotka-Volterra model of predator-prey dynamics [Lotka, 1910]. We show that both Transformers and RNNs can in context learn such complex dynamics, and thus provide preliminary but concrete evidence on the potential for *counterfactual story generation* that is important to scientific discovery and AI safety.

## Acknowledgements

MM thanks Jonas Peters for valuable discussions on the concept of counterfactual *reasoning* and the role of noise abduction. MM also thanks Hsiao-Ru Pan and Florent Draye for their input on identifiability and causal probing. MM acknowledges financial support from the *Konrad-Adenauer-Stiftung*.

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

## A  Posterior predictive distribution

*Proof of posterior predictive distribution.* Let $\Theta \subset \mathbb{R}, \mathrm{B} \subseteq \mathbb{R}$ and $\beta \in \mathrm{B}$. Define the observational data $\mathbf{x} = (x_1, y_1, ..., x_n, y_n)$ for fixed $n$. Then, with query $:= (x_1, y_1, ..., x_n, y_n, z, x^{\mathrm{CF}})$,

$$
\begin{aligned}
&p_{Y_Z^{\mathrm{CF}}}(y_z^{\mathrm{CF}}|\text{query}) \\
&= \int_\Theta p_{Y_Z^{\mathrm{CF}}}(y_z^{\mathrm{CF}}|\text{query}, \theta)\pi(\theta|\text{query})\mathrm{d}\theta \\
&= \int_\Theta p_{Y_Z^{\mathrm{CF}}}(y_z^{\mathrm{CF}}|\mathbf{x}, \theta, x^{\mathrm{CF}}, z)\pi(\theta|\mathbf{x}, x^{\mathrm{CF}}, z)\mathrm{d}\theta \\
&\stackrel{\perp\!\!\!\perp}{=} \int_\Theta p_{Y_Z^{\mathrm{CF}}}(y_z^{\mathrm{CF}}|\mathbf{x}, \theta, x^{\mathrm{CF}}, z)\pi(\theta|\mathbf{x})\mathrm{d}\theta && (8) \\
&= \int_\Theta \int_\mathrm{B} p_{Y_Z^{\mathrm{CF}}}(\beta(x^{\mathrm{CF}} - x_z) + y_z|\mathbf{x}, \theta, x^{\mathrm{CF}}, z, \beta)p_\beta(\beta|\mathbf{x}, \theta, x^{\mathrm{CF}}, z)\pi(\theta|\mathbf{x})\mathrm{d}\beta\mathrm{d}\theta \\
&\stackrel{\mathrm{dF}}{=} \int_\Theta \int_\mathrm{B} p_{Y_Z^{\mathrm{CF}}}(\beta(x^{\mathrm{CF}} - x_z) + y_z|x_z, y_z, \theta, x^{\mathrm{CF}}, z, \beta)p_\beta(\beta|\mathbf{x}, \theta)\pi(\theta|\mathbf{x})\mathrm{d}\beta\mathrm{d}\theta && (9) \\
&= \int_\Theta \int_\mathrm{B} \delta(Y_Z^{\mathrm{CF}} = \beta x^{\mathrm{CF}} + y_z - \beta x_z)p_\beta(\beta|\mathbf{x}, \theta)\pi(\theta|\mathbf{x})\mathrm{d}\beta\mathrm{d}\theta,
\end{aligned}
$$

where (8) follows from independence, as $\theta \perp\!\!\!\perp (X^{\mathrm{CF}}, Z)$, and (9) by de Finetti [de Finetti, 1931, Klenke, 2008], as each sequence $\{(x_1, y_1), ..., (x_n, y_n)\}$ forms an exchangeable unit. □

*Mean and variance in linear additive framework.* Both follow immediately by the tower law ($\mathbb{H}$) of iterated expectations,

$$
\begin{aligned}
\mathbb{E}[Y^{\mathrm{CF}}] &= \mathbb{E}[\beta X^{\mathrm{CF}} + U_Y] \\
&\stackrel{\mathbb{H}}{=} \mathbb{E}\left[\mathbb{E}\left[\beta|\theta\right]\right]\mathbb{E}[X^{\mathrm{CF}}] + \mathbb{E}\left[\mathbb{E}\left[U_Y|\theta\right]\right] \\
&= \mathbb{E}[\theta] = 0 \\
\mathrm{Var}(Y^{\mathrm{CF}}) &= \mathbb{E}[(Y^{\mathrm{CF}})^2] = \mathbb{E}\left[\left(\beta X^{\mathrm{CF}} + U_Y\right)^2\right] \\
&= \mathbb{E}\left[\left(\beta X^{\mathrm{CF}}\right)^2 + 2\beta X^{\mathrm{CF}}U_Y + U_Y^2\right] \\
&\stackrel{\perp\!\!\!\perp}{=} \mathbb{E}\left[\beta^2\right]\mathbb{E}\left[(X^{\mathrm{CF}})^2\right] + 2\mathbb{E}\left[\beta U_Y\right]\mathbb{E}\left[X^{\mathrm{CF}}\right] + \mathbb{E}\left[U_Y^2\right] \\
&\stackrel{\mathbb{H}}{=} \mathbb{E}\left[\mathbb{E}\left[\beta^2|\theta\right]\right]\mathrm{Var}\left(X^{\mathrm{CF}}\right) + \mathbb{E}\left[\mathbb{E}\left[U_Y^2|\theta\right]\right] \\
&= \mathbb{E}\left[1 + \theta^2\right]\left(\mathrm{Var}\left(X^{\mathrm{CF}}\right) + 1\right) \\
&= (1 + \mathrm{Var}(\theta))(12 + 1) = 13^2 = 169.
\end{aligned}
$$

The proof for the variance under the multiplicative extension, $\mathrm{Var}(\beta X^{\mathrm{CF}} U_Y)$, is analogous. □

## B  Transformation lemma and identifiability

### B.1  Transformation lemma

*Proof of Lemma 1.* As $T$ is invertible in $u$ given $f(x)$,

$$
y^{\mathrm{CF}} = T(f(x^{\mathrm{CF}}), u) = T\left(f(x^{\mathrm{CF}}), T^{-1}(f(x), y)\right).
$$

We require injectivity in $u$ to uniquely determine $u$ from $(f(x), y)$. Else, $y^{\mathrm{CF}} = T(f(x^{\mathrm{CF}}), u)$ is ambiguous and counterfactuals are ill-defined. Thus, the counterfactual completion writes

$$
Y^{\mathrm{CF}} = h\left(f(x^{\mathrm{CF}}), f(x), y\right)
$$

for $h : \mathcal{Y} \times \mathcal{Y} \times \mathcal{Y} \longrightarrow \mathcal{Y}$. □

## B.2 Counterfactual identifiability of bijective causal models

To prove Theorem 1, we restate Definition 6.1, Proposition 6.2, Lemma B.2 and Theorem 5.1 from Nasr-Esfahany et al. [2023]. For brevity, the proofs are omitted. We begin by introducing the *Bijective Generation Mechanism (BGM)* in our notation. Let $\{X_1, ..., X_d\}$ be a sequence of $d$ endogenous random variables. Each $X_j, j \in [d]$ is generated by

$$X_j := f_j\left(\text{Pa}(X_j), U_j\right).$$

Importantly, $f_j$ is a bijective mapping from $U_j$ to $X_j$ for each realization of $\text{Pa}(X_j)$ with $\text{Pa}(X_j)$ denoting the causal parents of $X_j$. In the bivariate case, we write

$$Y := f\left(\text{Pa}(Y), U_Y\right) = f(X, U_Y)$$

for $f$ bijective, as described in Lemma 1. Thus, with causal graph $X \longrightarrow Y$, we have $\text{Pa}(Y) = \{X\}, \text{Pa}(X) = \emptyset$.

**Definition 2** (Definition 6.1, Equivalence, Nasr-Esfahany et al. [2023]). BGMs $f_1$ and $f_2$ are equivalent iff there exists an invertible function $g(\cdot)$ such that

$$\forall x, y : f_1^{-1}(x, y) = g\left(f_2^{-1}(x, y)\right) \text{ or alternatively}$$

$$\forall x, u : f_1(x, u) = f_2\left(x, g^{-1}(u)\right).$$

**Proposition 1** (Proposition 6.2, Nasr-Esfahany et al. [2023]). BGMs $f_1, f_2$ produce the same counterfactuals $\iff f_1, f_2$ are equivalent BGMs.

**Lemma 2** (Lemma B.2, Nasr-Esfahany et al. [2023]). BGMs $f$ and $\hat{f}$ that produce the same distribution $P_{\mathcal{D}}(X, Y)$ are equivalent if

1. (Markovian) $U \perp\!\!\!\perp X$ and $\hat{U} \perp\!\!\!\perp X$.

2. for all $x$, $f(x, \cdot)$ and $\hat{f}(x, \cdot)$ are either strictly increasing or strictly decreasing functions.

**Theorem 2** (Theorem 5.1, Nasr-Esfahany et al. [2023]). *BGM $f$ is counterfactually identifiable given $\mathbb{P}_{X,Y}$ if*

1. *(Markovian) $U \perp\!\!\!\perp X$.*

2. *for all $x$, $f(x, \cdot)$ is either a strictly increasing or a strictly decreasing function.*

## B.3 Prove counterfactual identifiability under exchangeability

**Theorem** (Counterfactual identifiability under exchangeability, Theorem 1). Let $X, Y, U, \theta$ scalar random variables with $X \perp\!\!\!\perp U|\theta$. Take $T : \mathcal{Y} \times \mathcal{U} \longrightarrow \mathcal{Y}$ with $y = T(f(x), u)$. Assume $\forall f(x) \in \mathcal{Y}$, the inverse $T^{-1}(f(x), \cdot)$ exists for all $y$, i.e., $u = T^{-1}(f(x), y)$, and $\forall f(x) \in \mathcal{Y}$, $T(f(x), \cdot)$ is continuous. Suppose further that $f(x)$ continuous $\forall x$, and strictly monotonic in $x$. Then, set $Y = T(f(X), U)$. Given the joint law $\mathbb{P}_{X,Y|\theta}$, $T$ is counterfactually identifiable.

*Proof of Theorem 1.* We apply Lemma 2 and Theorem 2 [Nasr-Esfahany et al., 2023] to the case of conditional independence. The proof is analogous by taking $\mathbb{Q}_{X,Y} := \mathbb{P}_{X,Y|\theta}$. Clearly, the joint satisfies Markov conditional on the latent $\theta$ by design. Next by the inverse function theorem, $T(f(x), \cdot)$ is strictly monotonic, $\forall f(x)$, satisfying condition 2 in Theorem 2. With $f(x)$ strictly monotonic, continuous, $T'(x, \cdot) := T(f(x), \cdot)$ satisfies condition 2 for $x$. $\qquad\square$

# C Training details

To start, we introduce notation $[\cdot] := \{1, ..., \cdot\}$, for instance, $[n] := \{1, ..., n\}$. We construct synthetic datasets with fixed noise. Conditional on a uniformly distributed latent parameter $\boldsymbol{\theta}_i \in \Theta$, we draw $n_i \sim \mathcal{U}(\{2, ..., 50\})$ observational data points with noise $\mathbf{U}_X^{i;j}|\boldsymbol{\theta}_i, \mathbf{U}_Y^{i;j}|\boldsymbol{\theta}_i \in \mathbb{R}^E, j \in [n_i]$ and weights $\boldsymbol{\beta}_i|\boldsymbol{\theta}_i \in \mathbb{R}^E, i \in [N \cdot B]$ to enforce *non-i.i.d.* data. We choose $E = 5$. Next, we sample $N \cdot B$ data points with

$$
\begin{aligned}
\boldsymbol{\theta} &\sim \mathcal{U}([-6,6]^E) \quad &\boldsymbol{\beta}|\boldsymbol{\theta} &\sim \mathcal{N}(\boldsymbol{\theta}, \mathbf{I}_E) \\
\mathbf{U}|\boldsymbol{\theta} &\sim \mathcal{N}(\boldsymbol{\theta}, \mathbf{I}_E) \quad &\mathbf{X}^{\text{CF}} &\sim \mathcal{U}([-6,6]^E)
\end{aligned}
\tag{10}
$$

for $N = 50'000$ training steps of batch size $B = 64$. We thus have sample size $N \cdot B$. We set

$$\mathbf{Y} = \boldsymbol{\beta} \odot \mathbf{U}_X + \mathbf{U}_Y$$

with $\odot$ denoting element-wise products to have input and output embedding dimension agree without zero-padding. Including the indicator token $Z_b = z_b \cdot \mathbf{1}_E, z_b \in \{1, ..., n_b\}$, the corpus of queries then consists of $N$ batches of the form

$$\left\{ \left( \mathbf{x}_{b;1}, \mathbf{y}_{b;1}, ..., \mathbf{x}_{b;n_b}, \mathbf{y}_{b;n_b}, \mathbf{z}_b, \mathbf{x}_b^{\mathrm{CF}} \right) \right\}_{b \in [B]}$$

on which the model has to predict $\mathbf{y}_{b;z_b}^{\mathrm{CF}}, b \in [B]$. Our target $Y^{\mathrm{CF}}$ is zero-mean with $\mathrm{Var}(Y^{\mathrm{CF}}) = 169$ and $\log\left(\mathrm{Var}(Y^{\mathrm{CF}})\right) = 5.1299$.

We train on minimizing the per-batch mean squared error (MSE) between the counterfactual prediction $\widehat{\mathbf{y}_{[B]}^{\mathrm{CF}}}$ and the ground truth $\mathbf{y}_{[B]}^{\mathrm{CF}}$,

$$\mathrm{MSE}\left( \widehat{\mathbf{y}_{[B]}^{\mathrm{CF}}}, \mathbf{y}_{[B]}^{\mathrm{CF}} \right) = \frac{1}{B \cdot E} \sum_{b=1}^{B} \left\| \widehat{\mathbf{y}_{b;z_b}^{\mathrm{CF}}} - \mathbf{y}_{b;z_b}^{\mathrm{CF}} \right\|_2^2$$

and evaluate on an unseen test set following (10).

Our code is based on the repository by Garg et al. [2022]. We therefore adopt the learning rate of $10^{-4}$ for all function classes and models but use the AdamW optimizer [Loshchilov and Hutter, 2019] instead of Adam [Kingma and Ba, 2015]. We implement the experiments in `pytorch` [Paszke et al., 2019] and use one NVIDIA GeForce RTX 3090 GPU for training. The conducted experiments require between 5 minutes and 2 hours of training depending on model setup and task complexity.

# D   Model details

The Transformer architecture is described in Section 2.2. Here we lay out the details on the three recurrent models used and investigate the relevance of model depth in Transformers more closely.

## D.1   Model details on RNN architectures

The Elman RNN [Elman, 1990] is a foundational recurrent neural network architecture. It consists of an input layer, a hidden layer with recurrent connections, and an output layer. The hidden state at time step $t$, denoted by $h_t$, is computed as

$$h_t = \tanh(W_{ih}x_t + W_{hh}h_{t-1} + b_h)$$
$$y_t = W_{ho}h_t + b_o,$$

where $x_t$ is the input at time $t$, and $\tanh$ denotes the *tangens hyperbolicus*, a non-linear activation function. The output is given by $y_t$ and $W_., b_.$ represent learnable weight matrices and biases.

The Long Short-Term Memory (LSTM) network [Hochreiter and Schmidhuber, 1997] addresses the vanishing gradient problem by incorporating a memory cell and three gates: input $(i_t)$, forget $(f_t)$, and output $(o_t)$. These gates regulate the flow of information and enable the network to retain long-term dependencies. The LSTM cell is defined by the following equations,

$$f_t = \sigma(W_f x_t + U_f h_{t-1} + b_f)$$
$$i_t = \sigma(W_i x_t + U_i h_{t-1} + b_i)$$
$$o_t = \sigma(W_o x_t + U_o h_{t-1} + b_o)$$
$$g_t = \tanh(W_g x_t + U_g h_{t-1} + b_g)$$
$$c_t = f_t \odot c_{t-1} + i_t \odot g_t$$
$$h_t = o_t \odot \tanh(c_t).$$

In this formulation, $c_t$ is the cell state and $h_t$ the hidden output state. Activation vectors of the three gates are given by $i_t, f_t, o_t$ and $g_t$ is the cell input activation vector. By $\sigma$, we denote the sigmoid function and element-wise multiplication by $\odot$. $U_.$ represents another weight matrix.

The Gated Recurrent Unit (GRU) [Cho et al., 2014] introduces gating mechanisms to better control the flow of information. It simplifies the LSTM architecture by combining the forget and input gates into a single update gate, $z_t$. The GRU computes its hidden state using the following equations,

$$r_t = \sigma(W_r x_t + U_r h_{t-1} + b_z)$$
$$z_t = \sigma(W_z x_t + U_z h_{t-1} + b_r)$$
$$n_t = \tanh(W_n x_t + U_n(r_t \odot h_{t-1}) + b_n)$$
$$h_t = (1 - z_t) \odot n_t + z_t \odot h_{t-1}.$$

Here, $r_t$ is the reset gate, $z_t$ the update gate, $n_t$ the new gate, the candidate update vector. Note that the dependence of the hidden state on the update gate varies across documentations. We follow the definition in the `pytorch` package.

We perform a hyperparameter sweep on hidden size $D$ and the number of layers $L$ over the grid $D \in \{64, 128, 256\}, L \in \{1, 2\}$. We select the configuration with hidden size $256$ and $2$ layers for each of the three architectures.

### D.2   Copying in RNN architectures

In general, we study whether neural sequence models are able to perform counterfactual prediction. We observe that Transformers do so, but also RNN-type sequence models achieve low error. Figure 2 displays this finding for STANDARD GPT-2 Transformers as well as LSTMs, GRUs, and Elman RNNs. In their seminal paper, Hochreiter and Schmidhuber [1997] lay out the copy memory task for their developed method, the LSTM. Arjovsky et al. [2016] address vanishing and exploding gradients by modifying RNNs and Henaff et al. [2016] consider LSTMs and RNNs under longer contexts. Finally, Feng et al. [2024] test parallelizable modifications of LSTMs and GRUs on the Selective Copying Task [Gu and Dao, 2024]. Given this extensive literature, we restrict most of our analysis to autoregressive Transformers, as these represent the state-of-the-art architecture.

### D.3   Varying depth for additional values

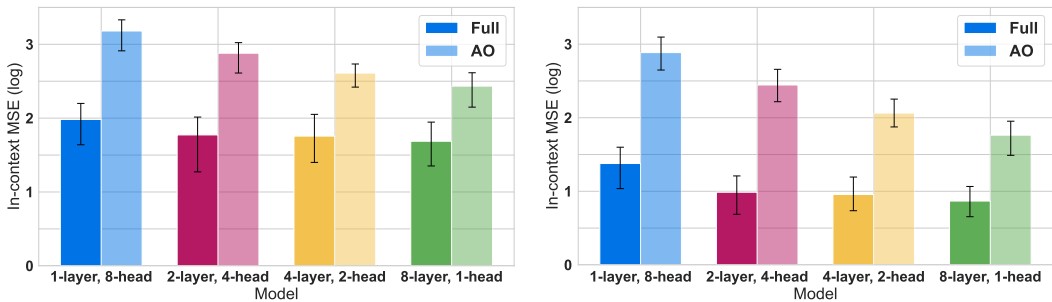

(a) **Varying depth for two in-context examples.**     (b) **Varying depth for forty-five in-context examples.**

Figure 7: **Varying depth more closely.** Building on top of the results displayed in Figure 3b, we compare **Full** and **AO** Transformers at a constant number of $8$ attention heads. We observe a decrease in in-context loss as model depth increases across shorter and longer contexts of $2$ and $45$ in-context examples, respectively. We evaluate on $6400$ sequences.

Analogous to Figure 3b, we note that increasing model depth leads to declining in-context MSE. This holds for contexts both shorter and longer than the $35$-example version considered above. Figure 7 illustrates the in-context MSE for varying depth, evaluated at $2$ and $45$ in-context examples, respectively. We note that the loss of the $1$-layer, $8$-head **AO** Transformer exhibits no significant differences between $2$ in-context examples and $45$. We interpret this as indication that the model is unable to infer information on the latent $\theta$ from the context. Reinforcing our findings that the $8$-layer, $1$-head Transformers achieves the lowest in-context MSE, this pattern serves as evidence that in-context inference occurs across layers. A contrasting notion may be that the Transformer maps different junks of information to different subspaces of the embedding space. Acting on each subspace individually, each attention head would then focus on a different task before the MLP would

combine the information and output the final prediction. Instead, the model appears to benefit from the interplay between layers, which supports the claim that attention heads across layers communicate through the residual stream [Elhage et al., 2021].

# E    Data diversity and robustness

## E.1    Data diversity adjustments and out-of-distribution

In Section 4.3, we discuss data diversity and generalizability to *OOD*. The UNIFORM setup largely follows the overall setup described in (10) with the following adjustment: instead of $N \cdot B \cdot E$, we sample $d \ll N$ realizations of the latent and compute the effective support size [Grendar, 2006]. Note that we control the absolute number of one-dimensional realizations as our setup effectively analyzes $E$ independent examples at every turn. Thus, we draw $d$ realizations and allocate across iterations and embedding dimensions. The parameter $d$ corresponds to the `diversity` argument in our code.

The NORMAL setup follows the above structure. Compared to (10), we sample the latent from $\boldsymbol{\theta} \sim \mathcal{N}(\mathbf{0}, \sqrt{12} \cdot \mathbf{I}_E)$ such that the theoretical variance is equivalent across setups. Everything else is analogous. Note that the effective support size is equivalent to the support size under the UNIFORM setup. This can be seen by interpreting Ess as the logarithm of the Shannon entropy [Shannon, 1948]. Indeed, $p(\vartheta)$ is uniformly distributed over discrete $\Theta_0$ if $\vartheta$ denotes a realization of the latent $\theta$,

$$H_{\text{UNIFORM}}(\theta) = - \sum_{\vartheta \in \Theta_0} \log p(\vartheta).$$

Therefore, we allocate the $d = |\Theta_0|$ realizations uniformly across iterations and embedding dimensions. Under the NORMAL setup, $p(\vartheta)$ depends on the distance of $\vartheta$ to 0. Here, we sample $d$ Gaussian realizations. We then allocate them proportional to $p(\vartheta)$ across iterations and embedding dimensions. In analogy to Figure 5a, we plot the in-context MSE against Ess for pre-training on the NORMAL setting and evaluating both in-distribution and on UNIFORM. Figure 8 illustrates the in-context MSE relative to effective support size on contexts of two examples. Here we train all models for $50'000$ training steps. Although the loss is at a higher level than for 35 in-context examples, we observe that the MSE declines with increasing effective support size. Moreover, when evaluated under the NORMAL setup, the model converges more rapidly than under the UNIFORM evaluation scheme. At $d = 3$, the Transformer already achieves performance comparable to that obtained when training on 1000 unique realizations. For the latter case, the results remain consistent with the findings above, indicating that an effective pre-training support size of approximately 10 is sufficient.

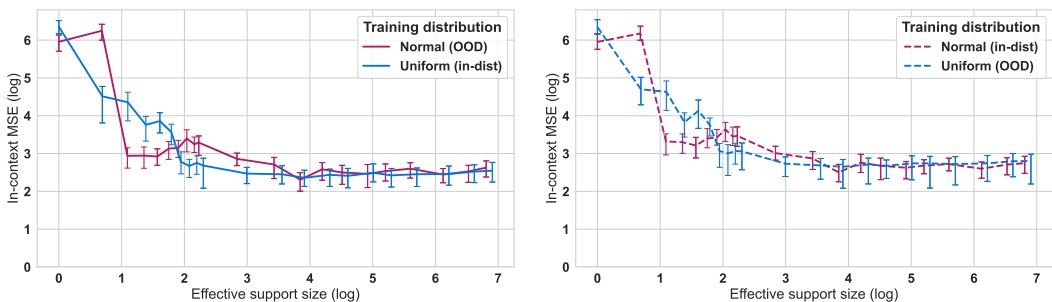

(a) **Data diversity for evaluation on UNIFORM.**      (b) **Data diversity for evaluation on NORMAL.**

Figure 8: **Data diversity for pre-training on two in-context examples.** We measure in-context MSE (log-scaled) averaged over $6400$ prompts at two in-context examples against log-scaled effective support size. Each point represents one fully pre-trained model on either the UNIFORM sampled or NORMAL sampled $\theta$. We train the models for $50'000$ training steps. All models are evaluated on both datasets. Error bars are the $95\%$ basic bootstrap confidence intervals.

## E.2    Non-linear and non-additive extensions

On top of the linear regression setting, we extend in-context counterfactual reasoning to non-linear, non-additive functions which are invertible in $u$. The general dataset setup remains the same with

adjustments made only to the computation of $y, y^{\text{CF}}$. We evaluate the 8-layer, 1-head **Full** and **AO** Transformers. Table 1 reports the in-context MSE on 6400 queries with 35 in-context examples each. Depending on the exact configuration of architecture and extension, the in-context MSE is at least one tenth of the empirical variance of the test set.

|  | $f(x, u)$ | Empirical variance | MSE of **AO** Transformer | MSE of **Full** Transformer |
|---|---|---|---|---|
| Tanh | $\tanh\left(\tau(\beta x + u)\right)$ | 0.3404 | 0.0132 | 0.0058 |
| Sigmoid ($\sigma$) | $\frac{1}{1+\exp(-\tau(\beta x + u))}$ | 0.0387 | 0.0019 | 0.0009 |
| Multiplicative | $\frac{1}{\sqrt{\text{Var}(\beta X^{\text{CF}} U_Y)}}\beta x u$ | 0.9274 | 0.0668 | 0.0364 |

Table 1: **Robustness to different function classes for regression tasks.** We report in-context MSE averaged over 6400 sequences for the 8-layer **AO** and **Full** Transformers under invertible nonlinear activation functions and multiplicative noise. We compute the empirical variance of target $Y^{\text{CF}}$ on the test set. We observe for both architectures that the MSE is 10 to 60 times lower than the empirical variance, suggesting both can in-context learn more complex function classes beyond linear regression and complete the counterfactual query. All functions are applied element-wise in one dimension. $\tanh$ and $\sigma$ are scaled by $\tau = \frac{1}{13}$ to counteract congestion around $\{-1, 1\}$ and $\{0, 1\}$, respectively. To guarantee theoretical variance of 1, we divide the multiplicative term by $\sqrt{\text{Var}(\beta X^{\text{CF}} U_Y)} = \sqrt{3410.4}$.

Note that under multiplicative noise, counterfactual reasoning reduces to a pure copying task as

$$Y^{\text{CF}} = \frac{1}{\sqrt{\text{Var}(Y^{\text{CF}})}}\beta X^{\text{CF}} U_Y = \frac{1}{\sqrt{\text{Var}(Y^{\text{CF}})}}\beta X^{\text{CF}}\frac{Y\sqrt{\text{Var}(Y^{\text{CF}})}}{\beta X} = \frac{X^{\text{CF}}}{X}Y,$$

where estimation of $\beta$ is not required for counterfactual completion.

### E.3 Data diversity in the literature

Recent theoretical research on data diversity aligns closely with our observations. Xiao et al. [2025] develop a formal framework showing that diversity in example selection improves in-context inference. With respect to model depth, Petty et al. [2024] show that at least two attention layers are required to support compositional generalization beyond memorization. This is in line with our findings. Figures 3b and 7b indicate that the additional benefit of deeper models decreases after having trained models of depth 2. Improvements are especially high for 2 layers relative to 1. Intuitively, we relate this to the multi-step process of performing counterfactual prediction. The model retrieves the relevant pair, $(x_j, y_j)$ for $z = j$, infers noise, and predicts the counterfactual $y^{\text{CF}}$ given $x^{\text{CF}}$ (Figure 1).

### E.4 Emergent learning of variable $z$ token

We confirm in Figure 4 that the model is capable of learning the latent parameter $\theta$ in context. We therefore find evidence that the model can learn the prior $\pi$ over the latent information as hypothesized in (2). The final part toward in-context counterfactual reasoning pertains *noise abduction*. Our setup is special as we include an embedding token $z$ which indicates the relevant in-context pair $(x_z, y_z)$ to abduct noise from. Only if the model can learn this connection on the fly, we can guarantee the three-step process of noise abduction, intervention, prediction. Indeed, we find that the **Full** GPT-2 Transformer with 12 layers and 8 heads trained for $1'000'000$ steps on $E = 1$ implements a *noise abduction head* at the 7th layer. We define a noise abduction head as one which attends to the relevant input-output pair $(x_z, y_z)$ when processing the $z$ token. This noise abduction head emerges at the sixth head of the seventh layer, and it attends to $y_z$ for variable realization of $z$. Visually inspecting the attention behavior of 100 batches averaged over 64 in-context sequences, we find that this arises in every single batch. Note that the $z$ token is constant within one batch. In Figure 9 we show the behavior for four different $z$ indices. For swift comparability, each prompt consists of 50 in-context examples. We conclude that noise abduction, central to counterfactual reasoning, emerges in a designated attention head.

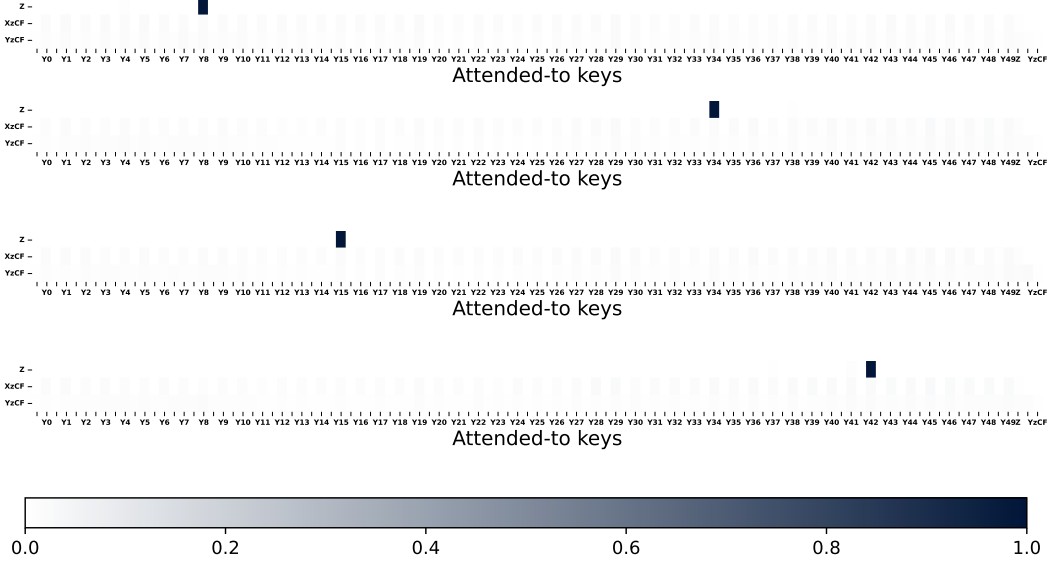

Figure 9: **Noise abduction head emerges in GPT-2 Transformer.** We plot 4 representative noise abduction heads evaluated on sequences with $z = 8, z = 34, z = 15, z = 42$, respectively. Each graph represents the average over $64$ sequences with identical $z$ index and 50 in-context pairs. We find that when processing embedding token $Z$, the model adapts to different values of $z$.

## E.5  Comparison to non-counterfactual regression

After reviewer feedback, we include a brief comparison of our results with training models on an observational continuation of the regression setup. Linear regression, as studied by Garg et al. [2022], does not require the ability to perform noise abduction. We take one step further by requiring the model to learn the functional form $f$ in context and to abduct the unobserved noise. To underscore our point, we train a **Full** GPT-2 architecture on $1'000'000$ training steps. Figure 10a represents the relative performance improvement of the counterfactual sequence relative to the observational setup. We find that for over 5 in-context examples the counterfactual MSE lies significantly below the estimated MSE of the observational sequence. In Figure 10b we observe the phase transition to a significantly lower in-context training MSE occurs after $300'000$ steps. This is in line with our hypothesis that in-context counterfactual reasoning, in principle, leads to a lower prediction error than the standard regression setup [Garg et al., 2022] as the noise is fixed and can be inferred directly.

# F  Model details for cyclic sequential dynamical systems

## F.1  Lotka-Volterra model

The Lotka-Volterra framework models the temporal evolution of the concentrations of two interacting species: predators and prey. In the absence of predators, the prey concentration grows continuously over time. Conversely, the predator concentration declines unless sustained by consuming prey. Predation enables predator growth, which simultaneously reduces prey concentration. We model the four individual dynamics by real, nonnegative parameters. The dynamics of the prey concentration (6) is parameterized by the intrinsic growth rate $\alpha$ and the rate at which the prey concentration decreases with predator concentration, $\beta$. For predator concentration (7), we capture the species-inherent decreasing concentration by $\gamma$ and the rate at which predators feed off prey by $\delta$. In the context of biological species, *concentration* can be thought of as the population density.

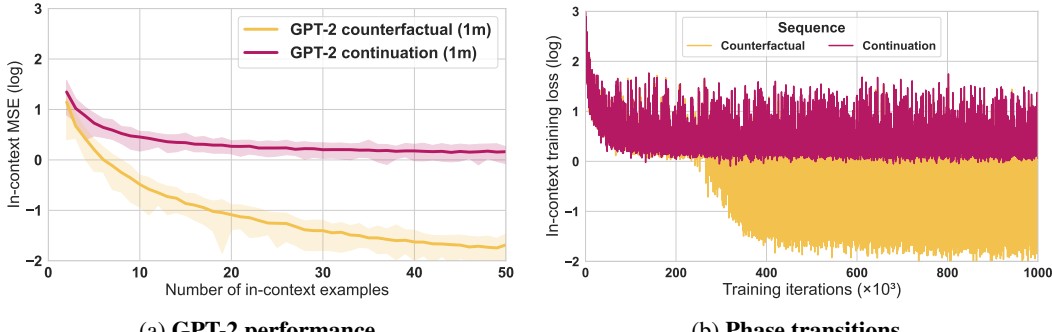

(a) **GPT-2 performance.**    (b) **Phase transitions.**

Figure 10: **Distinction between noise abduction and observational continuation.** We measure in-context MSE (log-scaled) averaged over $6400$ prompts for the **Full** GPT-2 Transformer. We observe significant improvements for contexts longer than $5$ examples. This we understand as indication of effective noise abduction in the model. We train the model for $1'000'000$ training steps and observe that the phase transition for the model trained on counterfactuals occurs after over $300'000$ steps. The logarithmic in-context training loss for the continuation framework remains at a higher level of above $0$.

### F.2 Bounds on Lotka-Volterra parameters

Similar to the regression setup, we draw the model parameters conditional on the latent $\theta$. In order to overcome exploding concentrations, we adopt the variation-of-constants method to bound the parameters by controlling the maximum and minimum concentrations. This ODE-based approach we utilize as a guiding principle throughout. As stochasticity may lead to a case where the bounds are violated, however, we *a posteriori* ensure evaluation on positive concentrations only. In the following, we adopt the ODE notation and write (6) and (7) with lower-case variables as

$$\frac{\mathrm{d}x_t}{\mathrm{d}t} = \alpha x_t - \beta x_t y_t, \tag{11}$$

$$\frac{\mathrm{d}y_t}{\mathrm{d}t} = -\gamma y_t + \delta x_t y_t \tag{12}$$

and observe that

$$\frac{\mathrm{d}x_t}{\mathrm{d}t} - (\alpha - \beta y_t)x_t = 0,$$

$$\frac{\mathrm{d}y_t}{\mathrm{d}t} - (-\gamma + \delta x_t)y_t = 0.$$

For now, we confine ourselves to (11). Dividing by $x_t$, we obtain

$$\frac{\mathrm{d}x_t/\mathrm{d}t}{x_t} - (\alpha - \beta y_t) = 0 \tag{13}$$

with $\frac{\mathrm{d}x_t/\mathrm{d}t}{x_t} = \frac{\mathrm{d}}{\mathrm{d}t}\log(x_t)$. By the fundamental theorem of calculus, integrating (13) over $[0, t]$ yields

$$\int_0^t (\alpha - \beta y_s)\,\mathrm{d}s = \int_0^t \frac{\mathrm{d}}{\mathrm{d}s}\log(x_s)\mathrm{d}s = \log(x_t) - \log(x_0).$$

By similar reasoning the result follows for (12) such that we can reorganize

$$x_t = x_0 \exp\left(\alpha t - \beta \int_0^t y_s \mathrm{d}s\right), \tag{14}$$

$$y_t = y_0 \exp\left(-\gamma t + \delta \int_0^t x_s \mathrm{d}s\right). \tag{15}$$

Now, we set $(\underline{x}, \bar{x}), (\underline{y}, \bar{y})$ to be lower and upper bounds of $x_t, y_t$, respectively, $t \in [0, T]$. Then,

$$\alpha t - \beta \underline{y} t \geq \log\left(\frac{x_t}{x_0}\right) \geq \alpha t - \beta \bar{y} t$$

such that

$$\alpha \geq \frac{1}{t} \log \left( \frac{x_t}{x_0} \right) + \beta \underline{y}, \quad \frac{1}{t} \log \left( \frac{x_t}{x_0} \right) + \beta \bar{y} \geq \alpha.$$

As $\log \left( \frac{\bar{x}}{x_0} \right) \geq \log \left( \frac{x_t}{x_0} \right) \geq \log \left( \frac{\underline{x}}{x_0} \right), \forall t$,

$$\alpha \in \left[ \frac{1}{t} \log \left( \frac{\bar{x}}{x_0} \right) + \beta \underline{y}, \frac{1}{t} \log \left( \frac{\underline{x}}{x_0} \right) + \beta \bar{y} \right].$$

For $t \in [0, T]$, the bound for $\lim_{t \searrow 0} \frac{1}{t}$ is trivial. We therefore decide to bound the final value at $T$ in $[\underline{x}, \bar{x}]$ such that

$$\alpha \in \left[ \frac{1}{T} \log \left( \frac{\bar{x}}{x_0} \right) + \beta \underline{y}, \frac{1}{T} \log \left( \frac{\underline{x}}{x_0} \right) + \beta \bar{y} \right].$$

To guarantee that the two bounds are in fact lower and upper bound, we can bound $\beta$,

$$\frac{1}{T} \log \left( \frac{\bar{x}}{x_0} \right) + \beta \underline{y} \leq \frac{1}{T} \log \left( \frac{\underline{x}}{x_0} \right) + \beta \bar{y}$$

$$\frac{1}{T} \log \left( \frac{\bar{x}}{\underline{x}} \right) \leq \beta \left( \bar{y} - \underline{y} \right)$$

$$\frac{1}{T (\bar{y} - \underline{y})} \log \left( \frac{\bar{x}}{\underline{x}} \right) \leq \beta.$$

Note that stating an upper bound for $\beta$ violates the model assumption $\beta \geq 0$.

Similarly, we write for (15),

$$-\gamma t + \delta \underline{x} t \leq \log \left( \frac{y_t}{y_0} \right) \leq -\gamma t + \delta \bar{x} t$$

$$-\gamma + \delta \underline{x} \leq \frac{1}{t} \log \left( \frac{\underline{y}}{y_0} \right) \leq \frac{1}{t} \log \left( \frac{\bar{y}}{y_0} \right) \leq -\gamma + \delta \bar{x}$$

such that for $T > 0$,

$$\gamma \in \left[ \delta \underline{x} - \frac{1}{T} \log \left( \frac{\underline{y}}{y_0} \right), \delta \bar{x} - \frac{1}{T} \log \left( \frac{\bar{y}}{y_0} \right) \right].$$

We, too, establish the bound by enforcing that lower and upper bounds are in correct order, i.e.,

$$\delta \underline{x} - \frac{1}{T} \log \left( \frac{\underline{y}}{y_0} \right) \leq \delta \bar{x} - \frac{1}{T} \log \left( \frac{\bar{y}}{y_0} \right)$$

$$\frac{1}{T} \left[ \log \left( \frac{\bar{y}}{y_0} \right) - \log \left( \frac{\underline{y}}{y_0} \right) \right] \leq \delta \left( \bar{x} - \underline{x} \right)$$

$$\frac{1}{T (\bar{x} - \underline{x})} \log \left( \frac{\bar{y}}{\underline{y}} \right) \leq \delta.$$

Put together, we have the bounds

$$\underline{\alpha} := \frac{1}{T} \log \left( \frac{\bar{x}}{x_0} \right) + \beta \underline{y} \quad \leq \alpha \leq \frac{1}{T} \log \left( \frac{\underline{x}}{x_0} \right) + \beta \bar{y} \quad =: \bar{\alpha},$$

$$\underline{\beta} := \frac{1}{T (\bar{y} - \underline{y})} \log \left( \frac{\bar{x}}{\underline{x}} \right) \quad \leq \beta,$$

$$\underline{\gamma} := \delta \underline{x} - \frac{1}{T} \log \left( \frac{\underline{y}}{y_0} \right) \quad \leq \gamma \leq \delta \bar{x} - \frac{1}{T} \log \left( \frac{\bar{y}}{y_0} \right) \quad =: \bar{\gamma},$$

$$\underline{\delta} := \frac{1}{T (\bar{x} - \underline{x})} \log \left( \frac{\bar{y}}{\underline{y}} \right) \quad \leq \delta.$$

### F.3 SDE Parameterization

In accordance with our exchangeable setting, we construct the dataset with

$$\boldsymbol{\theta} \sim \mathcal{U}([1,2]^E), \quad \mathbf{U}|\boldsymbol{\theta} \sim \mathcal{N}(\boldsymbol{\theta}, \mathbf{I}_E), \quad \mathbf{X}_0^{\mathrm{CF}}, \mathbf{Y}_0^{\mathrm{CF}} \sim \mathcal{U}([1,2]^E) \tag{16}$$

and bounds on the variables $(\underline{x}, \bar{x}) = (\underline{y}, \bar{y}) = (0.5, 2)$. We parameterize the Lotka-Volterra model,

$$\boldsymbol{\beta} - \underline{\boldsymbol{\beta}} \sim \mathrm{Exp}(\boldsymbol{\theta}) \qquad \boldsymbol{\delta} - \underline{\boldsymbol{\delta}} \sim \mathrm{Exp}(\boldsymbol{\theta})$$
$$\boldsymbol{\alpha} \sim \mathcal{U}\left([\underline{\boldsymbol{\alpha}}, \bar{\boldsymbol{\alpha}}]\right) \qquad \boldsymbol{\gamma} \sim \mathcal{U}\left([\underline{\boldsymbol{\gamma}}, \bar{\boldsymbol{\gamma}}]\right).$$

Finally, we sample the initial condition $(\mathbf{X}_0, \mathbf{Y}_0)$ from a $\mathrm{Beta}\left(\frac{1}{2-\boldsymbol{\theta}}, 2\right)$ distribution. Recall that the one-dimensional $\mathrm{Beta}(\kappa, \lambda)$ on the measurable space $([0,1], \mathfrak{B}([0,1]))$ attains its theoretical mode at $\frac{\kappa-1}{\kappa+\lambda-2}$ for $\kappa, \lambda > 1$, $\mathfrak{B}$ the Borel $\sigma$-field. Here we deviate from the conventional parameterization $\mathrm{Beta}(\alpha, \beta)$ to avoid conflict with the Lotka–Volterra parameters. To enforce an initial condition depending on the latent concept, we choose $(\kappa, \lambda)$ such that the theoretical mode equals $\theta - 1$. Equip the measurable space $(\Theta, \mathfrak{B}(\Theta))$ with probability measure $\pi$ with $\mathrm{supp}(\pi) = [1, 2]$. This is analogous to (1). To align the support of the $\mathrm{Beta}$ law with that of $\pi$, we require

$$\theta - 1 = \frac{\kappa - 1}{\kappa + \lambda - 2}$$
$$(\theta - 1)(\kappa + \lambda - 2) = \kappa - 1$$
$$\theta(\lambda - 2) + 3 - \lambda = (2 - \theta)\kappa$$
$$\frac{\theta(\lambda - 2) - (\lambda - 3)}{2 - \theta} = \kappa > 1.$$

Now, $\lambda > 1$ holds automatically such that we can set arbitrary $\lambda > 1$ and parameterize $\kappa = \frac{\theta(\lambda-2)-(\lambda-3)}{2-\theta}$. For straightforward implementation, we choose $\lambda = 2$ such that $\kappa = \frac{1}{2-\theta}$. To conclude, the initial value problem as well as the Lotka-Volterra parameterization depend on the latent $\theta$.

### F.4 Training details

We sample $n = 20$ distinct time steps $t_m \sim \mathcal{U}\left([0, 0.5]\right), m \in [n]$. In addition, we parameterize the Lotka-Volterra bounds defined above with $T = 1$. We again consider $E$ independent sequences at once by stacking independent one-dimensional SDEs. Within one example, all SDEs are evaluated at the same distinct time steps. At any $i$, we then provide the observational sequence as

$$\left(\mathbf{x}_{t_1}, \mathbf{y}_{t_1}, ..., \mathbf{x}_{t_n}, \mathbf{y}_{t_n}, \mathbf{z}, \mathbf{x}_{t_1}^{\mathrm{CF}}, \mathbf{y}_{t_1}^{\mathrm{CF}}\right)$$

for $\mathbf{z}$ a counterfactual delimiter token, identical across $i$. Given the query, we ask the model to predict the *full* completion of the counterfactual sequence, $\mathrm{comp}_i := (\mathbf{x}_{t_2}^{\mathrm{CF}}, \mathbf{y}_{t_2}^{\mathrm{CF}}, ..., \mathbf{x}_{t_n}^{\mathrm{CF}}, \mathbf{y}_{t_n}^{\mathrm{CF}})$, autoregressively. We again evaluate the model on the per-batch MSE of the predicted completion, $\widehat{\mathrm{comp}}_{[B]}$, and the underlying ground truth sequence, $\mathrm{comp}_{[B]}$,

$$\mathrm{MSE}\left(\widehat{\mathrm{comp}}_{[B]}, \mathrm{comp}_{[B]}\right) = \frac{1}{B \cdot E} \sum_{b=1}^{B} \left\|\widehat{\mathrm{comp}}_b - \mathrm{comp}_b\right\|_2^2$$

and evaluate on an unseen test set following (16).

We train for $N = 50'000$ training steps at batch size $B = 8$ and numerically approximate the solutions to the SDEs in (4) using the Euler–Maruyama [Maruyama, 1955] scheme. The counterfactual sequence is generated analogously but by recycling the Brownian motion used for the observational sequence. The manifestation of this can be seen in Figure 6b for *observational* and *counterfactual* $y$. We inherit the `torchsde` package from Li et al. [2020] and follow Charleux et al. [2018] for the implementation of the Lotka-Volterra equations. All other specifications are similar to the regression setup described in Appendix C. For fast approximation of the SDEs, we generate 20 batches of size $2500 \cdot 8$ at once. Across one batch, $\{t_1, ..., t_n\}$ is constant. At $N = 50'000$ training steps and a resulting dataset sample size of $50'000 \cdot 8$, we hence train on $\frac{50'000}{2500} = 20$ distinct realizations of the time sequence $\{t_1, ..., t_n\}$. As this can be seen as curriculum learning, we do not fully adhere to the aforementioned notion of exchangeability. Due to the complexity of approximating SDEs, however,

we give priority to an efficient implementation and acknowledge the resulting shortcoming. Last, the Euler-Maruyama method is an SDE approximation on $N$ *equal* subintervals of width $\Delta t > 0$. Therefore, we also train models with equidistant time steps $\{0, t_1, ..., t_n\}$ for constant $n = 20$. For this setup, the observational sequence starts at $t_0 = 0$,

$$\left( \mathbf{x}_0, \mathbf{y}_0, ..., \mathbf{x}_{t_n}, \mathbf{y}_{t_n}, \mathbf{z}, \mathbf{x}_0^{\text{CF}}, \mathbf{y}_0^{\text{CF}} \right)$$

with completion $\text{comp} := \left( \mathbf{x}_{t_1}^{\text{CF}}, \mathbf{y}_{t_1}^{\text{CF}}, ..., \mathbf{x}_{t_n}^{\text{CF}}, \mathbf{y}_{t_n}^{\text{CF}} \right)$ and evaluation on $n \cdot 2$ tokens. Throughout our experiments, we state the differences between these two setups.

### F.5 Attention behavior

Figure 12 shows that the 8-layer, 1-head **AO** Transformer trained on $N$ equal subintervals performs counterfactual reasoning by repeatedly moving information forward. Averaging over 8 test sequences, we observe that the tokens of the counterfactual sequence attend to the position of the delimiter token $\mathbf{z}$ and the initial condition $\mathbf{x}_0^{\text{CF}}$ at the first layer. This is intensified by layers 2 and 4 with attention to the initial $\mathbf{x}_0^{\text{CF}}$. Note that we only plot attention values between tokens of the counterfactual sequence. The Transformer at the fifth layer implements an induction head [Olsson et al., 2022] that copies forward the information one step. Afterwards, attention head 6 implements decaying attention over prior positions. Maximum weight is here put on the current token, and progressively lower weights on all previous tokens relative to token distance. This design facilitates forward information propagation across tokens. Similar behavior can be observed for the first two layers with attention to tokens $\{\mathbf{x}_0^{\text{CF}}, ..., \mathbf{x}_n^{\text{CF}}\}$.

In accordance with the hypothesis that induction heads drive in-context learning [Elhage et al., 2021, Olsson et al., 2022], we notice this pattern for larger models, with multiple heads per layer, and including the MLP. For instance, the **AO** STANDARD Transformer contains five heads across the first six layers which mostly attend to the delimiter token $\mathbf{z}$. At the eighth and ninth layer, the model implements each one induction head and one 2-gram [Akyürek et al., 2024] head in parallel. For the **Full** STANDARD Transformer, we similarly observe five copying heads which move information forward at layers 8 and 9. When trained on variable sequence length, the 8-layer, 1-head **AO** Transformer also exhibits this pattern with an induction head at layer 7. Taken together, this provides more evidence that counterfactual reasoning emerges, at least partly, in the self-attention layer of the Transformer.

## G  Connection between synthetic setup and natural language

### G.1 Discrete regression setup

Due to the synthetic nature of our setup, the connection to natural language is not immediately apparent. To illustrate how unobserved noise $u$ can be interpreted linguistically, we return to the introductory example. Each pair $(x_j, y_j)$ can be thought of as a pair of sentences describing a patient's treatment and effect. For $j \in [n]$, $x_j$ captures the treatment patient $j$ receives, and $y_j$ provides information on the effect of that medication. After presenting the model with $n$ such in-context examples, we prompt it to predict the counterfactual effect had patient $j = z$ received a different treatment. The model therefore needs to disentangle the subject-specific noise $u_j$ from the observational pair $(x_j, y_j)$. Figure 1 visualizes this process in natural language.

In addition to the one-prompt understanding, we can interpret the exchangeable setup such that each sequence resembles a different health issue. While we analyze breast cancer above, this enables training on a diverse set of treatment-effect pairs in order to perform inference over an unseen clinical picture. Establishing such a principled machine can assist informed decisions in personalized medicine.

### G.2 Continuous SDE setup

The cyclic causal dependencies modeled by the SDEs in (4) align with the sequential structure of natural language. Each query can be interpreted as a factual narrative of length $t_N$. Given a hypothetical initial condition $(x_0^{\text{CF}}, y_0^{\text{CF}})$, we task the model with counterfactually completing the story while keeping the noise fixed. This noise may represent semantic variability as well as narrative-internal uncertainty. By doing so, the model can generate plausible counterfactual continuations conditioned on the observed prompt. This approach resonates with the work of Qin et al. [2019], who explore how language models can reason about alternative story outcomes through counterfactual perturbations to character actions or events.

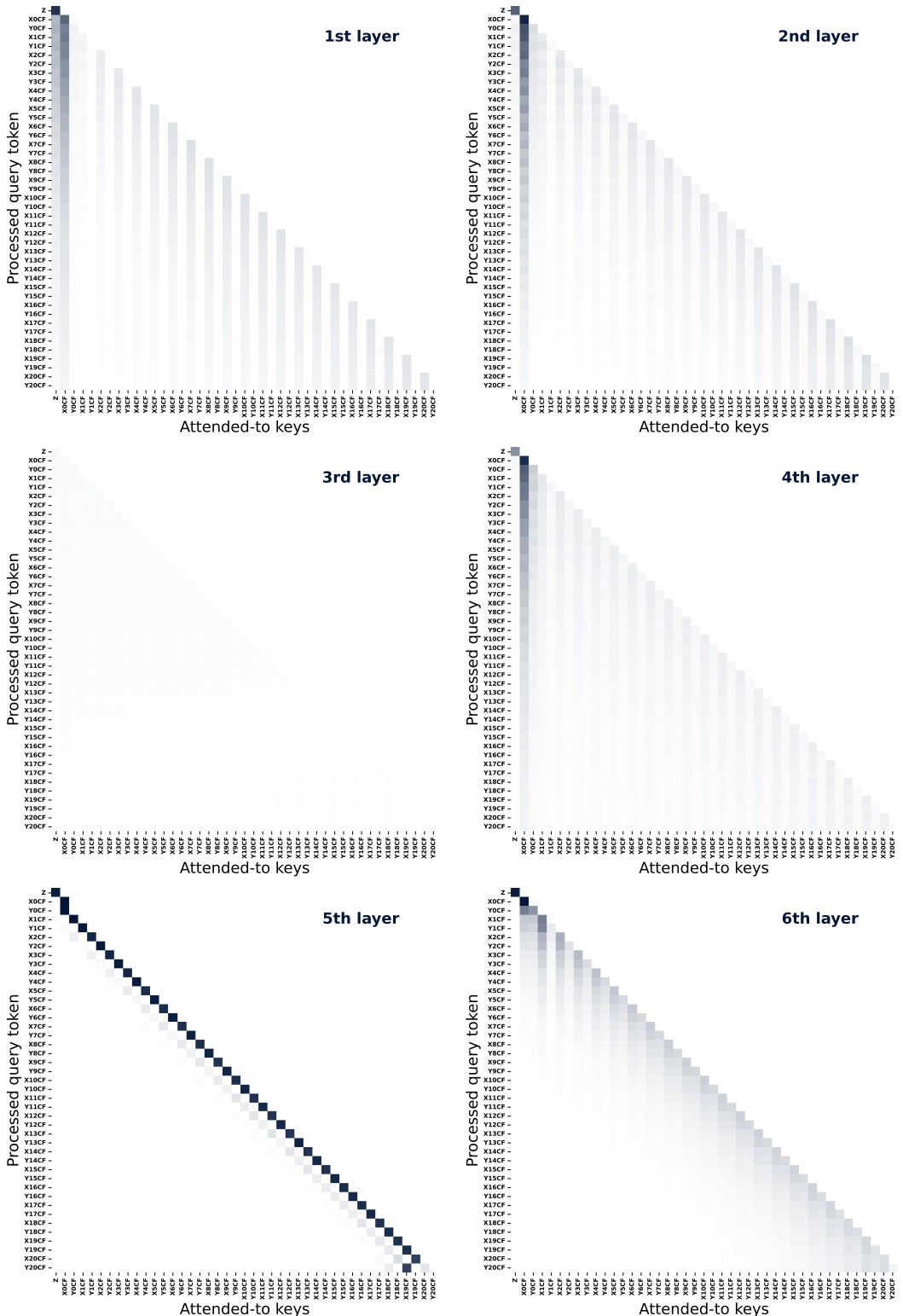

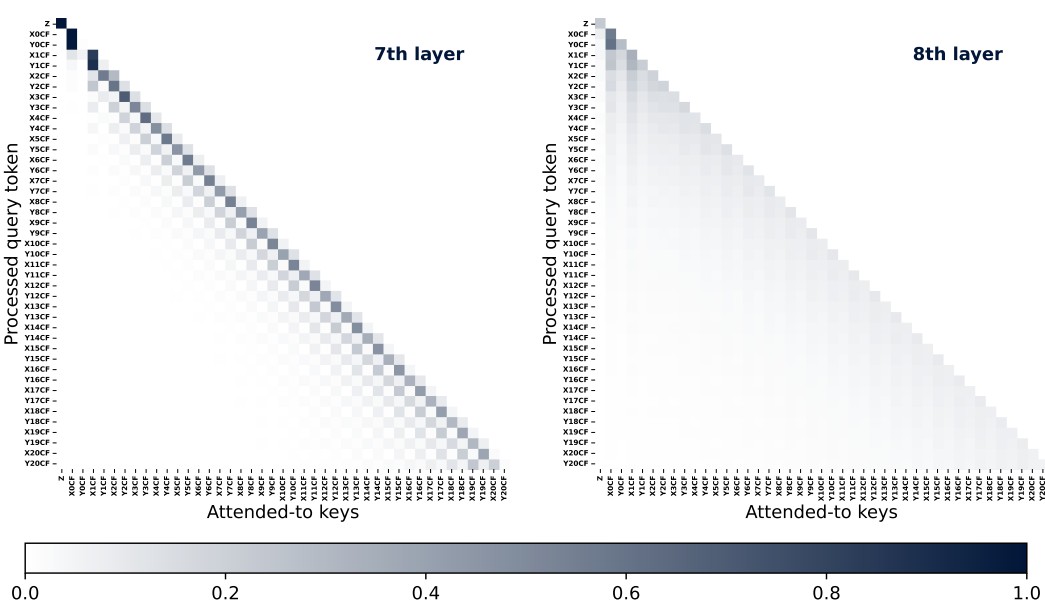

Figure 12: **Cyclic causal relationship.** The 8-layer **AO** Transformer, trained on counterfactual reasoning in Lotka-Volterra SDEs, includes two dedicated copying heads. At layers 5 and 7, induction heads shift residual stream information from the previous token forward one position. Earlier layers $1, 2$ and $4$ focus on the initial condition $\mathbf{x}_0^{\mathrm{CF}}$ and implement decaying attention over prior positions.

