# OpenReview forum: "Counterfactual reasoning: an analysis of in-context emergence"
_NeurIPS.cc/2025/Conference — NeurIPS 2025 poster_

### Official Review · Reviewer_nBNw · 2025-06-24

**Clarity:** 1
**Significance:** 2
**Originality:** 3
**Rating:** 3
**Confidence:** 4

**Summary:**

The authors present a study of counterfactual reasoning by in-context learning of large language models. They provide synthetic experiments that show, based on MSE performance, that language models are capable of performing the task.

**Questions:**

1. How does the example given in appendix G represent "what if" questions or explain the task of noise abduction in natural language? Please provide an instantiation of the variables used in the synthetic setup on a concrete example.
2. What is Figure 2 supposed to show: 1) Attention is required for optimal performance (GPT-2)? 2) Attention is not required since GRUs are a close second?
3. How would you evaluate the  "reasoning" capabilities of the model itself, not just the results?

**Ethical Concerns:**

["NO or VERY MINOR ethics concerns only"]

**Limitations:**

yes

**Paper Formatting Concerns:**

none found

**Quality:**

3

**Strengths And Weaknesses:**

Strengths:
- Rigorous theoretical underpinning of counterfactual reasoning
- Controllable study on synthetic data

Weaknesses:
- Due to the synthetic setup, it is very hard to understand what the experimental hypothesis is, and how the described experiment related to natural language generation. The example given in appendix G does not help since it does not represent "what if" questions nor does it explain the task of noise abduction in natural language.
- What is Figure 2 supposed to show: 1) Attention is required for optimal performance (GPT-2)? 2) Attention is not required since GRUs are a close second?
- The evaluation of the "reasoning" capabilities is based on the MSE results, not on an investigation of the reasoning process itself. It is well-known that the correctness of  the final conclusion does not necessarily reflect the validity of a model's reasoning process (see https://arxiv.org/abs/2404.01869, https://arxiv.org/abs/2305.18654)

---

> ### Author Rebuttal · Authors · 2025-07-31
>
> We thank the reviewer for the constructive feedback and are happy that the theoretical rigour of our work resonates with them. The feedback primarily centers around the presentation of results and definitions. We address the concerns below.
>
> W1 *[...] it is very hard to understand what the experimental hypothesis is [...]*
>
> **Our testable hypothesis is that neural sequence models are capable of performing counterfactual reasoning.**
>
> We agree that the conceptual leap from the synthetic setup to natural language is nontrivial. To restate our research question, we ask whether language models can perform in-context counterfactual reasoning (line 52). Today, most research on counterfactual story generation in large language models is based on prompting off-the-shelf language models [1]. To the best of our knowledge, there exists no prior work on rigorously disentangling the mechanism underlying counterfactual reasoning. An adequate toolkit to do so is causality [2]. Thus, we train a suite of neural sequence models on noise abduction. We supply a sequence of $n$ conditionally independent input-output pairs, $(x_k, y_k)$ for $k \in \{1,...,n\}$, where $y_k$ depends on $x_k$ and some noise $u_y^k$. Then, we ask the model to perform counterfactual prediction based on the pair $(x_j, y_j)$ for $j \in \{1,...,n\}$ by providing an index token $z = j$. The model is to infer the unobserved noise $u_y^j$ and to predict given the new intervention $x^\text{CF}$. Thus, the optimal prediction reads $y^\text{CF} = x^\text{CF} + u_y^j$.
> Here, we see the benefits of causality. The desired output $y^\text{CF}$ is unique and deterministic. In contrast to natural language, we can straightforwardly assess whether the model output coincides with the ground truth value. Below, we will argue how this also assists us in speaking about “reasoning.”
>
> Q1 *How does the example given in appendix G represent "what if" questions or explain the task of noise abduction in natural language?*
>
> **We provide a more flexible instantiation of our variables that is aligned with the problem setup.**
>
> We stress that the “what if” question is implicit in the model setup. By prompting the model with the relevant index token $z$ and the intervention $x^\text{CF}$, noise abduction occurs naturally. In the end, we will show how this idea opens up the opportunity to study noise abduction beyond natural language.
> We define “noise,” $u_y^k$, as the speaker’s personality or mood. The input-output pairs consist of a neutral event description, $x_k$, and the speaker’s subjective framing of that fact, $y_k$. Given a sequence of $n$ in-context examples sharing a common theme, $\theta$, we query the model to extract the noise from one of the examples, $(x_j, y_j)$ for $z = j$. Next, we supply a counterfactual query, $x^\text{CF}$ that is not based on the common theme, $\theta$. The model is then to copy the sentiment of the input-output pair at position $z$, and state the counterfactual continuation, $y^\text{CF}$. We give an instantiation below.
> Suppose the shared theme $\theta$ to be disruption events. Three input-output examples of neutral description and subjective continuation read:
>
> $x_1$ = [The train was late.], $y_1$ = [Well, I guess I’m walking again.]
>
> $x_2$ = [It started raining.], $y_2$ = [Yay, finally I have the chance to try out my new coat.]
>
> $x_3$ = [The meeting was canceled.], $y_3$ = [What a relief! I can prepare dinner early today.]
>
> Note the shared theme of disruption events. In addition, we observe three different moods ranging from (1) annoyed over (2) cheerful to (3) relieved. Consider now an intervention, $x^\text{CF}$, unrelated to the common $\theta$.
>
> $x^\text{CF}$ = [I found a slice of cake in the fridge.]
>
> Clearly, this is not a disruption event. Yet, depending on the index token $z \in \{1, 2, 3\}$ and the associated personality, we expect different continuations,
>
> $y^\text{CF} | z = 1$ = [Yet another person who didn’t finish their food. It’s still a shared space here.]
>
> $y^\text{CF} | z = 2$ = [Sweet! Today just keeps getting better.]
>
> $y^\text{CF} | z = 3$ = [Thank god, I don’t have to go buy dessert now.]
>
> Immediately, we see how the three personalities from above relate to the different continuations. In this bivariate setting, the “what-if” question is implicit. Instead of explicitly intervening on the examples in question format, we provide a new out-of-distribution example and the index the model should base its personality on.
> Whilst this may represent a carefully concocted illustration, we view in-context counterfactual reasoning as a general framework for latent attribute abduction and application. Beyond mood, this includes stylistic continuations across dialects, registers, or musical genres.
>
> Q2 *What is Figure 2 supposed to show: 1) Attention is required for optimal performance (GPT-2)? 2) Attention is not required since GRUs are a close second?*
>
> **Both classes can perform counterfactual reasoning, self-attention is the mechanism why Transformers can.**
> 1. We study whether neural sequence models are able to perform counterfactual prediction. We observe that Transformers and RNN-type sequence models do so (Figure 2). We stress that our RNN architectures have been tested on the (selective) copying task [3-6].
>
> 2. Given this, we confine ourselves to the analysis of autoregressive Transformers, as those represent the state-of-the-art architecture. Within the group of trained Transformers, we find that self-attention is relevant to perform counterfactual prediction. Given previous work on induction heads [7;8], we argue that this ability to abduct noise is attributed to counterfactual prediction [9] essentially involving the inference of unobserved noise from the observational pair $(x,y)$. In that sense, we deem self-attention relevant to counterfactual reasoning. Therefore, we argue that Transformers perform noise abduction in-context due to self-attention while RNN-type sequence models implement other mechanisms to copy information forward.
>
> Q3 *How would you evaluate the "reasoning" capabilities of the model itself, not just the results?*
>
> **We adopt the definition from causal literature and design our setup to interpret the MSE.**
>
> We share the reviewer’s concern that correct outputs do not guarantee valid reasoning processes. We list three arguments for choosing MSE.
> 1. We clarify our working definition of causal reasoning. Similar to [10], we define reasoning as “algebraically manipulating previously acquired knowledge to answer a new question.” In the context of causality, this reduces to “the process of drawing conclusions from a causal model” [11]. The latter statement also aligns with the work mentioned by the reviewer [12].
>
> 2. The reviewer states that evaluating the model on the final loss is insufficient [13]. This is correct if we consider complex Transformers which are trained on natural language. Especially when evaluating on integers, it is often not immediate whether the correct prediction can be attributed to legitimate reasoning. We do not disagree with that comment. As acknowledged in lines 50-53, this represents the very reason why we train a small Transformer under this controlled setup. We are able to invoke known results on the attention mechanism [7;8] while extending it to counterfactual prediction. By turning off the MLP sublayer for the most part, we exclude another source of complexity. With the 8-layer, 1-head attention-only Transformer we can track the steps (noise abduction, intervention, prediction) in the model. Put differently, we can understand the algebraic manipulation of the factual input $(x,y)$ in order to answer the new query $x^\text{CF}$. In accordance with the working definitions above, we deem the correct final prediction sufficient.
>
> 3. Recent research [9] has found that it is possible to perform counterfactual prediction. The setup we choose agrees with this modeling paradigm. To be explicit, we will append to our paper a corresponding theorem which relates that finding to our work. This implies that it is in principle possible to perform counterfactual reasoning simply from observing factual data $(x,y)$. Turning the question on its head, therefore, our analysis starts at the fact that counterfactual reasoning is theoretically possible. We then design a simple experiment which relates that finding to modern deep learning architectures. We turn off those sublayers which are not mechanistically interpretable. Finally, we confirm the hypothesis by invoking known results on how information is copied forward in self-attention.
>
> **Conclusion** We provide an instantiation of the variables used on a concrete example. Further, we clarify how our results on Transformers relate to other neural sequence models. Finally, we state our reasoning definition and connect it to existing work. We hope this addresses the reviewer's concerns. If our feedback resonates with the reviewer's perspective, we would like you to consider raising the score.
>
> **References**
>
> [1] Qin et al. (2019), Counterfactual Story Reasoning and Generation
> [2] Pearl (2009), Causality
> [3] Hochreiter et al. (1997), Long short-term memory
> [4] Arjovsky et al. (2015), Unitary Evolution Recurrent Neural Networks
> [5] Henaff et al. (2016), Recurrent Orthogonal Networks and Long-Memory Tasks
> [6] Feng et al. (2024), Were RNNs All We Needed?
> [7] Olsson et al. (2022), In-Context Learning and Induction Heads
> [8] Akyürek et al. (2024), In-Context Language Learning: Architectures and Algorithms
> [9] Nasr-Esfahany et al. (2023), On the Identifiability of Counterfactuals from Invertible Generative Models
> [10] Bottou (2011), From Machine Learning to Machine Reasoning
> [11] Peters et al. (2017), Elements of Causal Inference
> [12] Mondorf et al. (2024), Beyond Accuracy: Evaluating the Reasoning Behavior of Large Language Models
> [13] Dziri et al. (2023), Faith and Fate: Limits of Transformers on Compositionality

---

### Official Review · Reviewer_CTcD · 2025-07-01

**Clarity:** 3
**Significance:** 3
**Originality:** 3
**Rating:** 5
**Confidence:** 4

**Summary:**

The paper intends to answer the question whether LLMs can take a sequence of factual examples, imagine a “what-if” change, and generate the output without gradient updates. It examines the problem with a synthetic linear regression task requiring noise abduction.  A large transformer-based model is trained to fit the data from a mixture of sequences with different distributions but the same causal structure. Other models with different architectures, such as GRU and LSTM are trained on the same data for comparison. The paper concludes that large transformer models have the capability of in-context counterfactual reasoning.

**Questions:**

1. The paper assumes that a sequence of random variables is exchangeable. However, the order of words in an utterance matters in language model training. What is the implication of this assumption on LLM-generated counterfactuals?

2. Line 109-110 are not clear to me. How \theta is defined and how does it parameterise $U_X$ and $U_Y$?

**Ethical Concerns:**

["NO or VERY MINOR ethics concerns only"]

**Final Justification:**

The idea proposed in this paper is novel. The rebuttal addresses my concern on its technical treatment.

**Limitations:**

yes

**Quality:**

3

**Strengths And Weaknesses:**

S1: Examining the in-context counterfactual output generation capability of large transformers is an novel idea. This is an interesting direction considering the importance of counterfactual reasoning in decision making.

S2: A controlled study with linear regression is helpful to understand the transformer behaviour on counterfactual output generation.

W1: The training objective at line 116 minimises the counterfactual loss without considering the factual loss. Does the training have negative impact on factual output prediction? With such a single objective optimisation, the formulation is not much different to approaches studying emergence phenomenon of in-context learning itself, such as the following:

Garg, S., Tsipras, D., Liang, P.S. and Valiant, G., 2022. What can transformers learn in-context? a case study of simple function classes. Advances in Neural Information Processing Systems, 35, pp.30583-30598.

In a sense, the paper reaches a similar conclusion as the above paper due to that the training overfits counterfactual examples. I might have missed something here.

W2: In addition to transformers, the evaluation also shows results from other neural network architecture, e.g., GRU and LSTM also show counterfactual generation capability. I am wondering how the claim that counterfactual reasoning emerges in self-attention is justified.

W3: Line 41. it is better to describe \mu_y clearly here. In line 48, \mu_y is referred as context, but in Fig.1, it is referred as noise. Can you please also explain why the \mu_y is indexed by y not x?

---

> ### Author Rebuttal · Authors · 2025-07-31
>
> To begin with, we thank the reviewer for their constructive feedback and their positive opinion on our problem setup. Most of the concerns are related to notation and imprecise wording. We address the questions below.
>
> [W1] **We run new experiments and argue that the two setups are complementary, not identical.**
>
> We thank the reviewer for raising the concern on methodological innovation. We agree that the setup we study is comparable to that of [1]. This we openly acknowledge in our introduction (lines 40-41). We point out how our conclusions differ from that framework. We focus on noise abduction through transformations (Lemma 1). Linear regression, as studied by [1], does not require the ability to perform noise abduction. Thus, we design a problem setup that requires noise abduction for accurate prediction. In doing so, we build upon recent development in counterfactual identifiability [2]. Thus, we take one step further than [1]. Not only does our setup require the model to learn the functional form $f$ in-context, but also to abduct the unobserved noise. To the best of our knowledge, this is the first work that explicitly studies in-context noise abduction in neural sequence models.
> To underscore our point, we also trained an attention-only (AO) 8-layer, 1-head Transformer on the setup from [1] and found that its in-context MSE is 38% above the loss recorded for counterfactual prediction on the same model. We observe similar trends for Transformers of different configurations as well as RNN-type models. This is in line with our hypothesis that in-context counterfactual learning, in principle, leads to a lower prediction error than the standard regression setup [1] as the noise is fixed and can be inferred directly.
>
> In addition, we discuss potential negative impacts on factual output prediction. To do so, we perform a copying task. We take the 8-layer, 1-head attention-only Transformer and ask it to predict the factual completion $y_1$ of the input-output pair $(x_1, y_1)$ after seeing only that in-context example. As can be seen in Figure 3(a), the 8-layer AO architecture performs poorly on predicting counterfactuals  from one example only. Intuitively, this is in accordance with [1] as the functional form $f$ is to be learned in-context. The experiment we conduct here, however, is function-agnostic. We simply test whether the noise abduction as indexed by $z$ works as planned. We find that the in-context prediction of $y_1$ is 71% below the counterfactual prediction $y^\text{CF}$ with independently sampled $x^\text{CF}$. Taken together, we understand the two above results (on the setup from [1] relative to our design and on directly predicting $y_1$) as evidence that the model does not simply overfit on counterfactual examples. Instead, it appears to maintain an internal understanding of the three-step process: noise abduction through index token $z$, intervention based on $x^\text{CF}$, and prediction of $y^\text{CF}$.
> To summarize, our work is building on top of [1]. As it has been shown that prediction in a linear regression setting is possible [1], one can build a machine that combines the two paradigms. Our framework is flexible to include an index token $z = 0$ encoding linear regression. In that sense, we can understand $z$ as a switch for counterfactual prediction.
>
> [W2] **Both classes can perform counterfactual reasoning, self-attention is the mechanism why Transformers can.**
>
> First, we study whether neural sequence models are able to perform counterfactual prediction. We observe that Transformers do so, but also RNN-type sequence models achieve low error (Figure 2). We stress that our RNN architectures have been tested on the (selective) copying task [3;4;5;6]. Given the literature, we confine ourselves to the analysis of autoregressive Transformers, as those represent the state-of-the-art architecture. Hence, all future statements only refer to the Transformer. Within the group of trained Transformers, we find that self-attention is relevant to perform counterfactual prediction. Given previous work on induction heads [7;8], we argue that this ability to abduct noise is attributed to counterfactual prediction [2] essentially involving the inference of unobserved noise from the observational pair $(x, y)$. In that sense, we deem self-attention relevant to counterfactual reasoning.
>
> [W3] **We will unify the vocabulary in the next version.**
>
> We agree that the usage of the word “context” may be ambiguous here. We point the reviewer’s attention to the introduction. We understand counterfactual reasoning as the ability to predict the consequence of a hypothetical scenario, while keeping everything else constant. Put differently, we analyze the counterfactual output while fixing the context. In Pearl’s [9] causality framework, this is made explicit by defining a counterfactual statement as the prediction given fixed noise. Hence, by context, we refer to the noise in the former setting of counterfactual reasoning in natural language. It is correct that this may be misleading and we acknowledge that this subtlety is not made explicit in our work. Taking the reviewer’s advice seriously, we will unify the usage of “noise” and “context” throughout in the next version of the paper.
>
> Regarding your second point, we follow the standard notation for structural causal models [11]. Note that in the bivariate case, we define $X = U_X, Y = \beta X + U_Y$. Hence, we implicitly refer to the variable $X$ itself as the noise associated with variable $X$. This becomes clear if we consider more than two variables. Suppose we have $(X, Y, Z)$ with mutually independent noise $U = (U_X, U_Y, U_Z)$. Consider the case where both Y and Z have one causal parent, $X$. Here, $Y$ and $Z$ are independent when conditioning on $X$. Writing $Y = \beta X + U_X$ and $Z = \gamma X + U_X$, it is unclear how $X$ should be defined. More strikingly, $Y$ and $Z$ are not independent given $X$. We arrive at a contradiction. We hope that this example explains our choice of notation.
>
> [Q1] **We introduce the cyclic setup to capture sequentiality.**
>
> We apologise for the confusion and clarify. As laid out in lines 55-57, we assume exchangeability over input sequences, not within sequences. This is in line with current research [10] on relating in-context learning to De Finetti’s theorem. Within one input sequence, we assume that in-context examples are conditionally independent given the latent parameter, $\theta$. Having hopefully clarified this, we completely agree with the reviewer that “the order of words in an utterance matters.” This marks the very reason for our extensive analysis in section 5. Exactly because natural language is sequential and deeply interdependent (see lines 214-216), we introduce cyclic causality to our setup. Counterfactual story generation pertains to the completion of a hypothetical scenario after observing the factual narrative. Within one story, consecutive sentences depend on earlier ones. To model this sequential relationship but still maintain the benefits of evaluating on a synthetic setup, we utilize an SDE extension of the popular Lotka-Volterra [11] framework. That way we unite counterfactuals with the previous literature on in-context learning while drawing connections to the work on counterfactual story generation [12]. In doing so, we are able to circumvent the strong assumption of conditional independence across in-context examples.
>
> [Q2] **All noise is parameterised given the latent $\theta$.**
>
> Thank you for raising this point, we refer the reviewer to the more detailed setup in Appendix C (see line 143). In particular, we hope that equation (9) in the appendix.pdf file of the supplementary material can resolve the unclarity. To be explicit, we sample $\theta$ from a uniform distribution over the interval $[-6,6]$, and draw conditionally independent noise variables $U_X$ and $U_Y$ from a unit-variance Gaussian parameterised by mean $\theta$, i.e., $\mathcal{N}(\theta, 1)$. This way, we enforce conditional independence given the latent parameter $\theta$ both between $U_X$ and $U_Y$ and across in-context examples. Related to the reviewer’s previous question, we hope that this also clarifies the distinction between exchangeable input sequences and conditionally independent in-context examples.
>
> **Conclusion** To conclude, we thank the reviewer for their thoughtful feedback. We discuss the novelty of moving beyond i.i.d. data and our focus on noise abduction in neural sequence models. Our experimental comparisons further support that the model performs counterfactual reasoning rather than simple overfitting. We also spell out the reason for analyzing Transformers and explain the role of self-attention in abducting noise. Ambiguities in terminology (i.e., "context" vs. "noise") have been acknowledged, and we will revise the manuscript accordingly. Lastly, we have restated our assumptions about exchangeability and our data setup in Appendix C.
>
> **References**
> [1] Garg et al. (2022), What Can Transformers Learn In-Context?
> [2] Nasr-Esfahany et al. (2023), On the Identifiability of Counterfactuals from Invertible Generative Models
> [3] Hochreiter et al. (1997), Long short-term memory
> [4] Arjovsky et al. (2015), Unitary Evolution Recurrent Neural Networks
> [5] Henaff et al. (2016), Recurrent Orthogonal Networks and Long-Memory Tasks
> [6] Feng et al. (2024), Were RNNs All We Needed?
> [7] Olsson et al. (2022), In-Context Learning and Induction Heads
> [8] Akyürek et al. (2024), In-Context Language Learning: Architectures and Algorithms
> [9] Pearl (2009), Causality
> [10] Ye et al. (2024), Exchangeable Sequence Models Quantify Uncertainty Over Latent Concepts
> [11] Lotka (1910), Contribution to the Theory of Periodic Reactions
> [12] Qin et al. (2019), Counterfactual Story Reasoning and Generation

---

> > ### Comment · Reviewer_CTcD · 2025-08-05
> >
> > Thanks for the detailed response. One issue is that I cannot find paper [2] in the reference of the rebuttal. Is the title correct? Apart from this, the new experiment result on in-context mse addresses my main concern. I will increase my score accordingly.

---

> > > ### Author Response · Authors · 2025-08-05
> > > **Thank you for the thoughtful feedback!**
> > >
> > > We thank the reviewer for their constructive feedback on our paper. We are especially happy to hear that we were able to address the main concern. On that note, we also thank the reviewer for taking our points into consideration and raising the score. As we agree that the above represent relevant benchmarking results, we will include them in the next version of our paper.
> > > Last, we have indeed provided an incorrect title of [2]. We thank the reviewer for pointing this out. The correct source is of course, "Counterfactual Identifiability of Bijective Causal Models" by Nasr-Esfahany et al. (2023), published at ICML 2023.
> > > Once again, thank you to the reviewer!

---

### Official Review · Reviewer_oXZV · 2025-07-03

**Clarity:** 2
**Significance:** 3
**Originality:** 3
**Rating:** 4
**Confidence:** 3

**Summary:**

This paper studies in-context counterfactual reasoning using sequential models. It first introduces an in-context learning setup for counterfactual inference, aiming to answer queries of the form $P(y'_{x'} \mid x, y)$ given a causal graph $X \rightarrow Y$. The paper shows that this counterfactual query can be reduced to transformations on in-context observations. It then empirically demonstrates that transformers are capable of inferring such counterfactual queries in both linear and exchangeable sequence settings. Extensive empirical studies are also conducted to identify which components of training transformers or sequential models are critical for successful counterfactual inference. Finally, the experiments are extended to nonlinear and more dynamic sequential settings.

**Questions:**

**Q1.**
 Is there a more closely aligned example that directly illustrates the in-context counterfactual reasoning problem as formalized in the paper? (Please see the discussion in **W1**.)

**Q2.**
 Could the authors briefly connect the identifiability results in invertible structural causal models to Lemma 1? This connection seems highly relevant and, in my opinion, could strengthen the theoretical foundation of the paper.
 (See **W2** for further discussion.)

**Ethical Concerns:**

["NO or VERY MINOR ethics concerns only"]

**Final Justification:**

I will maintain my justification as previously communicated to the authors. I find the paper interesting, particularly in its modeling approach. It also validates certain known properties of transformers through experiments on a synthetic dataset. The authors’ rebuttal addressed my questions Q1 and Q2. However, the concerns raised in L_1 and L_2 remain unresolved. The paper neither provides theoretical insights into these properties nor validates them on larger datasets. Therefore, I recommend a borderline accept.

**Limitations:**

**L1.** The paper suggests that data diversity in pre-training, self-attention, and model depth are key for the performance of ctf reasoning with experiments. However, it lacks theoretical explanations or intuitive insights to support these observations. Including such a discussion would strengthen the paper and help guide future research.

**L2.** Additionally, as acknowledged by the authors, the experiments are conducted on small-scale synthetic datasets, which may limit the generalizability of the results. There could be a gap between these controlled settings and real-world applications.

**Quality:**

3

**Strengths And Weaknesses:**

# Strengths
S1. The paper is well-structured and clearly written. The problem statement and conclusions are presented clearly.

S2. The problem setting is well formalized and original, which may provide valuable insights for future work on in-context learning and counterfactual reasoning.

S3. The main conclusion, that language models can perform in-context counterfactual reasoning, is interesting and well supported by the experiments. This finding could guide future research in both causal inference and language modeling.


# Weaknesses

**W1. The example about John in the introduction (lines 30–35) is not well aligned with the actual problem setting of in-context counterfactual reasoning.** In the formal problem setup, the pairs $(x_i, y_i)$ are sampled from a family of SCMs sharing the same causal graph. However, the example, "John got the flu and is recovering in bed. Had he not attended the get-together, he would now visit his grandmother for cake.", is just a counterfactual statement and does not reflect the in-context formulation. It would be more helpful if the example directly illustrated the actual problem setting.

**W2. The theoretical insight (Lemma 1) it provides does not seem directly relevant to the core empirical claim of the paper.** The paper states that it "provides insights into how in-context counterfactual reasoning is equivalent to transformations on in-context observations (Lemma 1)," but this does not clearly support the conclusion that transformers are capable of performing counterfactual reasoning.

In particular, Lemma 1 suggests that $y^{\mathrm{CF}}$ can be expressed as a transformation of $x,y, x^{\mathrm{CF}}$ (Equation 2). However, this transformation may not be unique. If the true transformation underlying the data does not match the learned model's transformation, then the counterfactual prediction could be entirely incorrect. This corresponds to the identification issue mentioned later in the paper. Similar identifiability issues for invertible SCMs are discussed in prior work [1]. I believe combining such identification results with Lemma 1 would significantly strengthen the theoretical support for the paper’s main claim.

[1] Nasr-Esfahany, Arash, Mohammad Alizadeh, and Devavrat Shah. "Counterfactual identifiability of bijective causal models." International Conference on Machine Learning. PMLR, 2023.

PS: I am happy to raise the score if the concerns are solved but may lower score if the concerns are not solved.

---

> ### Author Rebuttal · Authors · 2025-07-31
>
> We thank the reviewer for their constructive feedback and for appreciating our experimental setup. The concerns are mostly of technical nature, which we address below.
> Q1 *Is there a more closely aligned example that directly illustrates the in-context counterfactual reasoning problem?*
> **We provide a more flexible instantiation of our variables that is aligned with the problem setup.**
> We agree that the conceptual leap from the example we use in the introduction to our problem setup is large. Consider the following.
> We define “noise,” $u_y^k$, as the speaker’s personality or mood. The input-output pairs consist of a neutral event description, $x_k$, and the speaker’s subjective framing of that fact, $y_k$. Given a sequence of $n$ in-context examples sharing a common theme, $\theta$, we query the model to extract the noise from one of the examples, $(x_j, y_j)$ for $z = j$. Next, we supply a counterfactual query, $x^\text{CF}$ that is not based on the common theme, $\theta$. The model is then to copy the sentiment of the input-output pair at position $z$, and state the counterfactual continuation, $y^\text{CF}$. We give an instantiation below.
> Suppose the shared theme $\theta$ to be disruption events. Three input-output examples of neutral description and subjective continuation read:
> $x_1$ = [The train was late.], $y_1$ = [Well, I guess I’m walking again.]
> $x_2$ = [It started raining.], $y_2$ = [Yay, finally I have the chance to try out my new coat.]
> $x_3$ = [The meeting was canceled.], $y_3$ = [What a relief! I can prepare dinner early today.]
> Note the shared theme of disruption events. In addition, we observe three different moods ranging from (1) annoyed over (2) cheerful to (3) relieved. Consider now an intervention, $x^\text{CF}$, unrelated to $\theta$.
> $x^\text{CF}$ = [I found a slice of cake in the fridge.]
> Clearly, this is not a disruption event. Yet, depending on the index token $z \in \{1, 2, 3\}$ and the associated personality, we expect different continuations,
> $y^\text{CF} | z = 1$ = [Yet another person who didn’t finish their food. It’s still a shared space here.]
> $y^\text{CF} | z = 2$ = [Sweet! Today just keeps getting better.]
> $y^\text{CF} | z = 3$ = [Thank god, I don’t have to go buy dessert now.]
> Immediately, we see how the three personalities from above relate to the different continuations. In this bivariate setting, the “what-if” question is implicit. Instead of explicitly intervening on the examples in question format, we provide a new out-of-distribution example and the index the model should base its personality on. Whilst this may represent a carefully concocted illustration, we view in-context counterfactual reasoning as a general framework for latent attribute abduction and application. Beyond mood, this includes stylistic continuations across dialects, registers, or musical genres.
> Q2 *Could the authors briefly connect the identifiability results in invertible structural causal models to Lemma 1?*
> **We introduce a new theorem that links the identifiability results under Markov to our exchangeable setup.**
> We understand the reviewer’s point that the connection between the counterfactual prediction involving a transformation on the observed values $x, y, x^\text{CF}$ and the Transformer effectively learning the causal structure is not immediate. First, we point out that the class of functions we consider can be subsumed under the class of bijective generation mechanisms [1]. Second, each function we test our hypothesis on (App. E.2) is monotonic and continuous on $\mathbb{R}$. Hence, by the inverse function theorem, they are invertible ($\Longleftrightarrow$ bijective). Last, invoking the exchangeable setup we use throughout, we have conditional independence of the form $X \perp U | \theta$. Our mechanism $T(f(x),\cdot)$ invertible, continuous (for $f(x)$ invertible, continuous) is then monotonic for all $f(x)$. Given the law $\mathbb{Q}{X,Y} := \mathbb{P}{X,Y,\theta}$, we invoke Lemma B.2 and Theorem 5.1 from [1] and claim:
>
> Let $X, Y, U, \theta$ scalar random variables with $X \perp U | \theta$. Take $T: \mathcal{Y} \times \mathcal{N} \longrightarrow \mathcal{Y}$ with $y = T(f(x),u)$. Assume $\forall f(x) \in \mathcal{Y}$, the inverse $T^{-1}(f(x),\cdot)$ exists for all $y$, i.e., $u = T^{-1}(f(x),y)$, and $\forall f(x) \in \mathcal{Y}$, $T(f(x), \cdot)$ is continuous. Suppose further that $f(x)$ continuous $\forall x$, and strictly monotonic in $x$. Then, set $Y = T(f(X),U)$. Given the joint law $\mathbb{P}_{X,Y|\theta}$, $T$ is counterfactually identifiable.
>
> The proof to this statement is analogous to [1]. Regarding efficient learning of the underlying causal structure, we invoke previous literature on the Transformer’s ability to learn various function classes in-context [2]. We will include this connection in the revised version.
> L1 *The paper suggests that data diversity in pre-training, self-attention, and model depth are key for the performance of ctf reasoning with experiments. [...] Including such a discussion would strengthen the paper and help guide future research.*
> **We provide prior literature to underscore our Transformer-related findings.**
> Below, we establish an intuitive connection. In the next version, we are happy to include a more detailed discussion.
> Recent theoretical research on data diversity aligns closely with our observations. [3] develop a formal framework showing that diversity in example selection improves in‑context inference. With respect to model depth, [4] show that at least two or three attention layers are required to support compositional generalization beyond memorization. This is in line with our findings. Figure 3(b) indicates that the additional benefit of deeper models decreases after having trained models of depth 2. Improvements are especially high for 2 layers relative to 1. Intuitively, we relate this to the multi-step process of performing counterfactual prediction. The model retrieves the relevant pair, $(x_j,y_j)$ for $z = j$, infers noise, and predicts the counterfactual $y^\text{CF}$ given $x^\text{CF}$ (Figure 1). Adopting the residual stream view introduced by [5], we argue that we require multiple layers to copy the information forward. We observe distinct attention patterns [6;7]. While a typical head in the first layer attends to the $z$-token, we also find heads copying the information of the counterfactual intervention forward. As noise abduction requires two steps - copying $x_j$ and $y_j$ can happen in parallel in different heads and transforming the latter to obtain $u_j^y$ occurs at the next layer - we intuitively relate our results to established findings on self-attention.
> L2 *There could be a gap between these controlled settings and real-world applications.*
> **We follow the literature in designing controlled experiments to confirm our theoretical hypothesis on a small scale.**
> We agree with the reviewer that real-world applications like natural language are likely to be more complex than the synthetic task we study. The main purpose of our paper is to connect the theoretical result of performing counterfactual prediction to known properties of Transformers.
> 1. We are interested in confirming that Transformers are capable of performing noise abduction while learning a certain functional form in-context. Showing this theoretically through Lemma 1 in combination with the above statement, we check whether this is in principle possible on a small scale.
> 2. We envision that the mechanism behind in-context counterfactual reasoning remains identical for larger models. Extrapolating results from smaller models is in line with existing literature on the theoretical understanding of large models. For instance, [9] implement a two-layer attention-only Transformer with neither biases nor layer normalization. Similarly, [8] extrapolate their findings from an architecture with linear self-attention. Even more so, our work is grounded in earlier findings on induction heads [6;7] and the in-context learning of functions $f$ [2], which, too, are derived from small models.
> 3. Synthetic tasks play a central role in causal inference. Controlled setups often form the backbone of key theoretical insights. [9] benchmark causal discovery methods on synthetic data before moving to real-world examples like the canonical temperature-altitude case. Similarly, [10] evaluate their Do Finetti algorithm exclusively on synthetic data. In our work, we aim to demonstrate that a causal mechanism can emerge in-context, without constructing a large, standalone model ourselves. Instead, we hope this work lays the groundwork for in-context counterfactual reasoning by integrating causality with in-context learning.
>
> **Conclusion** We thank the reviewer again for their thoughtful feedback and their refreshing input on counterfactual identifiability. We have added a more aligned example and formally connected the cited paper. If questions remain, feel free to reach out. Otherwise, we appreciate your consideration in raising the score as suggested.
> **References**
> [1] Nasr-Esfahany et al. (2023), On the Identifiability of Counterfactuals from Invertible Generative Models
> [2] Garg et al. (2022), What Can Transformers Learn In-Context?
> [3] Xiao et al. (2025), The Role of Diversity in In-Context Learning for Large Language Models
> [4] Petty et al. (2023), The Impact of Depth on Compositional Generalization in Transformer Language Models
> [5] Elhage et al. (2021), A Mathematical Framework for Transformer Circuits
> [6] Olsson et al. (2022), In-Context Learning and Induction Heads
> [7] Akyürek et al. (2024), In-Context Language Learning: Architectures and Algorithms
> [8] von Oswald et al. (2023), Transformers Learn In-Context by Gradient Descent
> [9] Peters et al. (2014), Causal Discovery with Continuous Additive Noise Models
> [10] Guo et al. (2024), Do Finetti: on Causal Effects for Exchangeable Data

---

> > ### Comment · Reviewer_oXZV · 2025-08-04
> >
> > Thanks for the detailed response from the authors. The text example provided for **Q1** is promising, and the theoretical analysis for **Q2** is sound. I think the text example will serve as a good illustration to connect the work to real-world settings.
> >
> > I understand that the purpose of the paper is to connect the current theoretical results to known properties of transformers. To accomplish this goal, I believe either (**L1**) adding more theoretical findings that directly relate to transformer properties, or (**L2**) including more experimental validation on complex datasets, would make the paper more compelling.
> > For this reason, I will keep my current score, as it is already a positive one. I look forward to seeing the final revision!

---

> > > ### Author Response · Authors · 2025-08-05
> > > **Thank you for your constructive feedback!**
> > >
> > > Thank you for the positive review. We are happy that the text examples and theoretical analysis resonate with the reviewer's perspective. Moreover, we appreciate your suggestions to strengthen the paper with Transformer-related theory and more experimental validation. Taking your input seriously, we will incorporate your thoughts into our next revision and look forward to the updated manuscript.

---

### Official Review · Reviewer_3LJY · 2025-07-03

**Clarity:** 3
**Significance:** 3
**Originality:** 3
**Rating:** 4
**Confidence:** 3

**Summary:**

The paper analyzes LLMs’ capabilities to perform in-context counterfactual reasoning. To this end, the authors prompt a model with what-if questions based on a numeric synthetic dataset using a function-learning probe. The authors focus on a linear SCM (Structural Causal Model) setting, working in the latent space, where they add noise to various stages of token representations.
In a GPT-2–sized Transformer, they show that counterfactual prediction copies the noise from the factual pair and applies a learned transformation. They empirically demonstrate that it uses fewer in-context examples than other architectures (e.g., LSTMs). The authors further show that performance degrades when self-attention is removed (using only the MLP), and that the mixture and number of SCMs influence pre-training. They argue that their findings also generalize to data mimicking counterfactual story generation.

**Questions:**

Have you conducted any preliminary experiments on real-world tasks to assess the generalizability of your approach?

Beyond analyzing attention scores, have you explored other interpretability techniques, such as methods from mechanistic interpretability (e.g., probing the residual stream)?

**Ethical Concerns:**

["NO or VERY MINOR ethics concerns only"]

**Final Justification:**

The authors provided detailed explanation on the theoretical approaches and also the comparison to existing work. While the contribution is mainly theoretical, the approach would benefit from evaluation on real-world examples and also on a larger scale. E.g., papers focusing on induction heads evaluate on real-world examples, such as on the IOI task [1], or [2] on factual knowledge; (I find the distinction between theoretical and method-oriented approaches not to be exclusive in recent interpretability research). Therefore, I will keep my original score that is leaning to accept the paper.

[1] https://arxiv.org/pdf/2211.00593 [2] https://arxiv.org/pdf/2405.17969

**Limitations:**

Yes

**Quality:**

3

**Strengths And Weaknesses:**

Strenghts:

The authors focus on a very timely topic: counterfactual reasoning. The paper is generally well-structured and well-written.

The task and synthetic dataset are well designed for the research question and thoroughly developed: the task is methodologically well-formalized, and the experiments follow this design closely, answering relevant questions and presenting meaningful ablation studies. In this way, the authors present a sound theoretical framework to test SCMs in an in-context counterfactual reasoning setting.

Through this approach, the authors provide several insights—such as the importance of self-attention—which constitute a meaningful contribution to the growing body of work on reasoning in LLMs.

Weaknesses:

Since the approach is only tested on a synthetic dataset, its general applicability to real-world tasks remains uncertain. It would have been valuable to include an evaluation on at least one real-world dataset.

Furthermore, the method is evaluated only on a comparatively small model.

The approach primarily builds on existing function-learning probing techniques, applying them to in-context counterfactual reasoning. Moreover, the task of answering "what-if" questions has been addressed in prior work. While the paper tackles a meaningful problem in a well-controlled setting, I find that the methodological novelty is rather modest.

---

> ### Author Rebuttal · Authors · 2025-07-30
>
> We thank the reviewer for their thoughtful feedback on our project. We are happy to hear that the reviewer finds our method methodologically well-formalized and the topic timely. The concerns raised are mostly related to model size and dataset setup. Below, we provide answers to the questions raised and discuss some of the mentioned weaknesses.
>
> W1 *Since the approach is only tested on a synthetic dataset, its general applicability to real-world tasks remains uncertain. It would have been valuable to include an evaluation on at least one real-world dataset. Have you conducted any preliminary experiments on real-world tasks to assess the generalizability of your approach?*
> **Our work  shows that noise abduction in neural sequence models is in principle possible.**
> The main purpose of our paper is to connect the theoretical result of performing counterfactual prediction to known properties of Transformers. Primarily, we are interested in confirming that Transformers are capable of performing noise abduction while learning a certain functional form in-context. Showing this theoretically through Lemma 1, we check whether this is in principle possible on a small scale. Note that is in contrast to a methods paper that proposes new algorithms.
>
> W2 *The method is evaluated only on a comparatively small model.*
> **We adopt the standard approach to design a small-scale experiment to underscore our theoretical findings.**
> For a paper trying to provide theoretical understanding, we think it is preferable to train as small a model as possible while capable of exhibiting the desired performance (in our case, in-context counterfactual reasoning). This is similar to previous literature where the analysis is restricted to small models. For instance, [1] implement a two-layer attention-only Transformer with neither biases nor layer normalization. Similarly, [2] extrapolate their findings from an architecture with linear self-attention. In addition, our work is grounded in earlier findings on induction heads [3;4] and the in-context learning of functions $f$ [5], which, too, are derived from small models. As we are interested in showing that the discovered mechanism arises in-context, we do not intend to build a counterfactual reasoning machine ourselves.
>
> W3 *The approach primarily builds on existing function-learning probing techniques, applying them to in-context counterfactual reasoning. Moreover, the task of answering "what-if" questions has been addressed in prior work. While the paper tackles a meaningful problem in a well-controlled setting, I find that the methodological novelty is rather modest.*
> **We introduce results from counterfactual identifiability to in-context learning and propose the idea to model natural language by stochastic differential equations.**
> To the best of our knowledge, existing work that addresses “What-if” questions focuses on direct evaluation of pre-trained large language models through question-answer pairs.  In lines 263-269, we refer to different lines of work that tackle the natural language continuation of hypothetical scenarios. However, such direct evaluation on pre-trained large models creates ambiguities: first, it is unclear how to evaluate the correct response in natural language; second, the uncontrolled sources in pre-training data leads to ambiguities in attributions of success. We thus opt to train neural sequence models from scratch, i.e., we do not rely on off-the-shelf foundation models to perform counterfactual reasoning. Further, we are the first who rigorously attempt to bridge the gap between recent work on counterfactual identifiability [6] and well-studied properties of Transformers.
>
> With respect to the theoretical analysis of Transformers, the closest to our work is [5]. We adopt a similar setting for our regression setup, but explicitly focus on the noise abduction process. On top of that, we study a cyclic SDE-based setup which, to the best of our knowledge, previous work did not perform. As language is inherently sequential, counterfactual story generation pertains to the completion of a hypothetical scenario after observing the factual narrative. Within one story, consecutive sentences depend on earlier ones. To model this sequential relationship but still maintain the benefits of evaluating on a synthetic setup, we utilize an SDE extension of the popular Lotka-Volterra [7] framework. That way we combine counterfactuals with the previous literature on in-context learning while drawing connections to the work on counterfactual story generation [8]. We are not aware of any other work that tested in-context learning in a counterfactual setup, where it is essential to abduct the noise. In that sense, our work closely aligns with the literature on counterfactual identifiability. To the best of our knowledge, we are the first ones to extend this theoretical finding on invertible, i.e., bijective, functions to Transformers and RNN-type models. In essence, we therefore lay the theoretical foundation for counterfactual reasoning in neural sequence models.
>
> All that being said, we highly value the feedback of our reviewer. That is, if our response has not addressed your concern, we would be happy if you could provide some references adjacent to our work. As we are not aware of any literature which rigorously tackles counterfactual reasoning in an in-context learning setting, we would be grateful to hear your thoughts.
>
> Q2 *Beyond analyzing attention scores, have you explored other interpretability techniques, such as methods from mechanistic interpretability (e.g., probing the residual stream)?*
> **We experimented with different techniques (including causal probing) and decided on attention maps due to their interpretable results and prevalence in the literature.**
> We thank the reviewer for suggesting probing techniques. When sketching the outline of this paper, we considered different techniques from mechanistic interpretability to include in our work. Apart from attention visualization, we thought about gradient-based explainability methods and probing. For several reasons, we have decided on presenting attention maps as our interpretability method of choice.
> To start off, the literature on in-context learning is centered around induction heads [1;3;4]. The standard way to analyse which head copies what information in attention-only Transformers is through attention maps. As we understand noise abduction as an implicit mechanism to copy the unobserved noise forward, we connect our findings to that line of work.
> On top of that, relative to other mechanistic interpretability methods, attention maps are straightforward to grasp. In an otherwise technical paper, we understand attention maps to be a refreshing oasis for readers from different communities.
> Most importantly, causal probing techniques have recently been put under scrutiny. First, several authors argue that seeing probes as evidence for causal properties may be misleading [9;10]. Others underline this idea by providing concrete examples [11]. Second, it is unclear to what degree probes can assist the engineer in explaining the underlying structure. One desideratum of probing is its low complexity [12]. If, for instance, we use nonlinear MLP-based probes, we might be able to predict some latent parameter [13]. Now, the MLP itself is not interpretable in the desired way. In this sense, we are able to “explain” the model by some mechanism. That mechanism itself we do not understand, however. Our initial goal of understanding the model shifts to a mechanism-understanding problem.
> All that being said, we performed linear probing on our baseline architecture: the 8-layer, 1-head attention-only Transformer. To do so, we fit a linear model to predict $u_y$ from the residual stream, $r$, after every sublayer. For each of the sublayers, we fit a weight matrix and perform all of the following experiments on the residual streams of all sublayers. For clarity, we have dropped indices. Based on the fitted weight matrix $\hat{W}$, we construct the projection matrix $P$ and predict $r’ = (I - P)r$. This idea of nullifying the linear information carried by the residual stream did not yield relevant results. Similarly, computing the Moore-Penrose pseudoinverse, $W^\dagger$, and injecting a fresh $u_y^\text{new}$ into the residual stream through $r’’ = r + W^\dagger \left( u_y^\text{new} - \hat{u}_y \right)$ for the linearly fitted noise $\hat{u}_y$ did not allow us to causally steer the prediction of $y^\text{CF}$. Taking a step back, this is in line with our intuition as the self-attention sublayer includes nonlinearities through the $\mathrm{softmax}$ and layer normalization. Therefore, we reaffirm our choice of interpretability method as attention maps are adopted by the community and straightforwardly decodable.
>
> **References**
> [1] Elhage et al. (2021), A Mathematical Framework for Transformer Circuits
> [2] von Oswald et al. (2023), Transformers Learn In-Context by Gradient Descent
> [3] Olsson et al. (2022), In-Context Learning and Induction Heads
> [4] Akyürek et al. (2024), In-Context Language Learning: Architectures and Algorithms
> [5] Garg et al. (2022), What Can Transformers Learn In-Context?
> [6] Nasr-Esfahany et al. (2023), "On the Identifiability of Counterfactuals from Invertible Generative Models."
> [7] Lotka (1910), Contribution to the Theory of Periodic Reactions
> [8] Qin et al. (2019), Counterfactual Story Reasoning and Generation
> [9] Elazar et al. (2020), Amnesic Probing: Behavioral Explanation with Amnesic Counterfactuals
> [10] Ravichander et al. (2020), Probing the probing paradigm: Does probing accuracy entail task relevance
> [11] Geiger et al. (2021), Causal Abstractions of Neural Networks
> [12] Alain et al. (2016), Understanding intermediate layers using linear classifier probes
> [13] Hewitt et al. (2019), Designing and Interpreting Probes with Control Tasks

---

> > ### Comment · Reviewer_3LJY · 2025-08-05
> > **Response to Authors**
> >
> > Thank you very much for the detailed response!
> > I appreciate the detailed explanation on the theoretical approaches and also the comparison to existing work. While I understand that the contribution is mainly theoretical, I still find that the approach would benefit from evaluation on real-world examples and also on a larger scale. Please see papers focusing on induction heads evaluate on real-world examples, such as on the IOI task [1], or [2] on factual knowledge; (I find the distinction between theoretical and method-oriented approaches not to be exclusive in recent interpretability research). Therefore, I will keep my original score that is leaning to accept the paper.
> >
> > [1] https://arxiv.org/pdf/2211.00593
> > [2] https://arxiv.org/pdf/2405.17969

---

> > > ### Author Response · Authors · 2025-08-05
> > > **Thank you for the constructive feedback**
> > >
> > > We thank the reviewer for their feedback and suggestions on additional literature. We find the papers interesting as they take a method-oriented approach and also analyse GPT2.
> > >
> > > Indeed, we agree that the distinction between the theoretical and method-oriented approaches need not necessarily be exclusive. In fact, the motivation of our paper is a thorough understanding through our rigorous theoretical analysis. We hope that it inspires a new method-oriented paper. As of right now, we think the theoretically motivated analysis may already be of interest to the community.
> > >
> > > Thank you again for your review and for engaging with our paper so thoughtfully. We will incorporate your suggestions into the next version.

---

### Decision · Program_Chairs · 2025-09-17

**Decision:**

Accept (poster)

**Comment:**

This paper evaluates the ability of LLMs to perform in-context counterfactual reasoning. The experiments explore the internal mechanisms supporting counterfactual reasoning in transformers (and find the architectural factors that support / hinder in-context counterfactual reasoning). All experiments are performed on a synthetic dataset. The experiments seem sound and comprehensive, but as mentioned by several reviewers, the framing (of, e.g., natural language counterfactual reasoning; the examples given in the rebuttal also don't seem to exhibit counterfactual reasoning, which is more like what-if-instead (i.e., conditioned on alternatives), but simply what-if reasoning) is very far from the actual problem being studied, and the discussion of "emergence" of reasoning is not implied by their ability to perform such reasoning when being trained on a task that implicitly requires it. I would suggest the authors significantly revise the framing and clarity in any final version of the paper.